# GENERATIVE LEARNING FOR SOLVING NON-CONVEX PROBLEM WITH MULTI-VALUED INPUT-SOLUTION MAPPING

**Enming Liang and Minghua Chen**[*]
School of Data Science, City University of Hong Kong

## ABSTRACT

By employing neural networks (NN) to learn input-solution mappings and passing a new input through the learned mapping to obtain a solution instantly, recent studies have shown remarkable speed improvements over iterative algorithms for solving optimization problems. Meanwhile, they also highlight methodological challenges to be addressed. In particular, general non-convex problems often present multiple optimal solutions for identical inputs, signifying a complex, multi-valued input-solution mapping. Conventional learning techniques, primarily tailored to learn single-valued mappings, struggle to train NNs to accurately decipher multi-valued ones, leading to inferior solutions. We address this fundamental issue by developing a generative learning approach using a rectified flow (RectFlow) model built upon ordinary differential equations. In contrast to learning input-solution mapping, we learn the mapping *from input to solution-distribution*, exploiting the universal approximation capability of the RectFlow model. Upon receiving a new input, we employ the trained RectFlow model to sample high-quality solutions from the input-dependent distribution it has learned. Our approach outperforms conceivable GAN and Diffusion models in terms of training stability and run-time complexity. We provide a detailed characterization of the optimality loss and runtime complexity associated with our generative approach. Simulation results for solving non-convex problems show that our method achieves significantly better solution optimality than recent NN schemes, with comparable feasibility and speedup performance.

## 1 INTRODUCTION

Constrained optimization problems have widespread applications in various engineering domains, including wireless communication, power systems, and autonomous driving. Conventional iterative methods (e.g., Gurobi and Mosek) have demonstrated satisfactory performance in solving optimization problems with static parameters. However, these methods fall short in real-time operational contexts, where problems must be solved repeatedly under rapidly changing contextual parameters. Examples of these applications include solving AC optimal power flow problems in real-time power grid operations, semi-definite programming-based real-time scheduling, and coding operations in modern wireless communication systems.

Recently, machine learning (ML) based strategies, including the end-to-end (E2E) solution mapping (Pan et al., 2019; 2020; Donti et al., 2021), learning-to-optimize (L2O) iterative scheme (Khalil et al., 2016; Heaton et al., 2020; Chen et al., 2021b), and hybrid approaches (Baker, 2019; Fioretto et al., 2020; Park & Van Hentenryck, 2023), have shown potential in tackling such real-time constrained optimization problems. The E2E approach, boosted by the universal approximation capabilities of neural networks (NN) (Hornik et al., 1989; Leshno et al., 1993), learns the mapping between the input contextual parameters and the optimal solutions. Given new input parameters, trained NNs can directly output the solution by forward calculation with considerable speedup compared to iterative solvers (Pan et al., 2019; Kotary et al., 2021a; Donti et al., 2021).

Besides ensuring the feasibility of solutions with respect to the input-dependent constraint set (Donti et al., 2021; Tabas & Zhang, 2022; Zhao et al., 2022; Liang et al., 2023), issues associated with

---

[*]Corresponding Author: Minghua Chen (`minghua.chen@cityu.edu.hk`).

multiple solutions arise for non-convex problems (Nandwani et al., 2020; Kotary et al., 2021a; Pan et al., 2023), where there exist multiple global optimum or symmetric solutions with the same objective value. The existing supervised training approaches using NN face challenges in learning this multi-valued input-solution mapping, as illustrated in Fig. 1.

To tackle the issues related to the multi-valued input-solution mapping, researchers have utilized various prior knowledge to either isolate one of the multiple input-solution mappings (Nandwani et al., 2020; Kotary et al., 2021a) or learn multiple mappings directly (Li et al., 2018; Courts & Kvinge, 2021; Pan et al., 2023). For instance, leveraging the prior knowledge that the solution mapping for optimization problems often satisfies a local Lipschitz condition, Kotary et al. (2021a) develops a training dataset generation scheme to circumvent the issue of multiple solutions. Pan et al. (2023) utilizes initial points of the deterministic solver to characterize different solution mappings and learn a single mapping augmented with input initial points. However, these approaches either rely on specific prior knowledge of the non-convex problem or do not ensure the stability of the training process and performance of predictions. More discussions on related works are in Sec. 2.

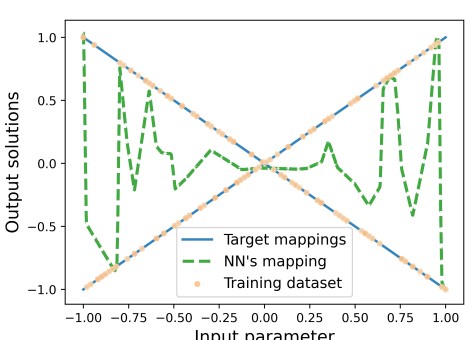

Figure 1: Consider the setting where a non-convex problem admits two optimal solutions with the same optimal objective value for each input, leading to a 2-valued input-solution mapping. An NN trained with 100 uniformly sampled input-solution pairs and MSE loss function fails to learn a legitimate mapping.

In this paper, instead of learning input-solution mappings as in prior arts, we employ Rectified Flow (RectFlow) (Liu et al., 2022) as a generative approach to learn the mapping from input to a preferred solution distribution for solving non-convex problems. We made the following contributions.

▷ In Sec. 3, we elucidate the challenge of solving non-convex problems characterized by multi-valued input-solution mapping. In response to this challenge, we propose a novel generative learning framework, detailed in Sec. 4. Our approach is predicated on the availability of a training dataset sampled from a preferred input-dependent solution distribution, and leverages the universal approximation capability of the RectFlow model to learn the distribution. Given new input, this trained model is subsequently harnessed to generate high-quality solutions from the learned distribution.

▷ In Sec. 5, we theoretically characterize the incurred optimality loss of generated solutions in Sec. 5. The results show that the probability that the generated solution incurs an optimality gap larger than any $\delta > 0$ decreases exponentially to zero as we sample more from the learned input-dependent solution distribution. We also characterize the runtime complexity and discuss the training complexity of the framework.

▷ In Sec. 6, simulation results for learning multi-valued mappings and solving various non-convex problems show that our method achieves significantly better solution optimality than the latest NN-based schemes, with comparable feasibility and speedup performance.

To our best knowledge, this is the first work advocating a paradigm shift from learning input-solution mapping to learning the mapping from input to solution-distribution for solving continuous constrained optimization problems. The code is available at GL_Code.

## 2 RELATED WORK

Recent advancements in ML-driven optimization, encompassing end-to-end (E2E) solution mapping, learning-to-optimize (L2O) strategies, and hybrid models, have demonstrated promise in addressing constrained optimization challenges (Kotary et al., 2021b). In the setting of supervised training, high-quality solutions are prepared in advance, and a neural network (NN) is trained to map an input parameter to the solution. However, for non-convex problems, the existence of multiple solutions leads to a multi-valued input-solution mapping, which introduces significant challenges when using NNs for approximation. We summarize existing proposals to address the challenges introduced by multi-valued mappings in Table 1.

Table 1: Existing studies for solving non-convex puzzles with a multi-valued input-solution mapping.

| Existing Studies (see Sec. 2 for references) | Stable training | Multiple solutions | Optimality guarantee | Low training complexity | Low run-time complexity |
|---|---|---|---|---|---|
| Data generation | ✓ | ✗ | ✓ | ✗ | ✓ |
| Data selection | ✗ | ✗ | ✗ | ✗ | ✓ |
| Data augmentation | ✓ | ✓ | ✓ | ✗ | ✓ |
| Hindsight loss | ✓ | ✓ | ✗ | ✓ | ✓ |
| Clustering approach | ✗ | ✓ | ✗ | ✓ | ✓ |
| **Generative Learning** | ✓ | ✓ | ✓ | ✓ | ✓ |

**Supervised learning for input-to-solution mappings**. To learn single input-solution mapping, Kotary et al. (2021a) utilizes the local Lipschitz condition to design a data generation scheme and avoid the multiple solution issues in the training dataset. Without preparing the single-solution dataset in advance, Nandwani et al. (2020) proposes a data selection approach, which trains another NN to identify one of multiple mappings from a dataset mixed with numerous optimal solutions for each input. To learn multiple mappings directly from the dataset, Li et al. (2018) adopts multiple models to divide one-to-many mappings and utilizes the hindsight loss to optimize them jointly. The clustering methods (Courts & Kvinge, 2021) can also be applied to partition the dataset and learn different mappings separately. Sun et al. (2017) and Pan et al. (2023) transform the multi-valued mapping into a single-valued one with the augmentation of initial points information from deterministic solvers. These methodologies leverage unique aspects of the solution dataset, yet they tend to depend on additional information, such as initial points, and often lack a performance guarantee.

**Other approaches for input-solution mappings**. In scenarios lacking ground-truth optimal solutions, unsupervised learning emerges as a viable alternative for learning an input-to-solution mapping under the guidance of a well-designed differentiable loss function (Pogančić et al., 2019; Karalias & Loukas, 2020; Huang & Chen, 2021; Qiu et al., 2022; Ferber et al., 2023). Reinforcement learning approaches can also be employed to train solution mappings for non-convex problems, particularly for combinatorial problems with discrete structures (Kwon et al., 2020; Zhang et al., 2020; Xin et al., 2021; Ma et al., 2021; Caramanis et al., 2024). However, these approaches can easily become stuck in local optima without guidance of ground-truth optimal information.

**Generative models for approximating distribution**. The multi-valued input-solution mapping can be reframed as an input-conditioned solution distribution from which multiple solutions are sampled. Leveraging the sequential nature of routing problems, auto-regressive (AR) distributions are trained to capture multiple symmetric solutions (Kool et al., 2018; Kwon et al., 2020; Xin et al., 2021; Hou et al., 2022). To model more intricate distributions, advanced generative models are brought into play. These models transform a simple prior distribution into the complex data distribution either via push-forward mappings or through the integration of stochastic/ordinary differential equations (SDE/ODE). The push-forward generative models, such as generative adversarial networks (GANs), are known to struggle with the mode collapse issue, which limits their ability to capture multi-modal distributions effectively (Salmona et al., 2022). In contrast, SDE/ODE-based generative models, like diffusion models, learn the continuous transformation dynamics rather than a single mapping, which has been demonstrated superior performance in various application scenarios (Ho et al., 2020; Song et al., 2020b; 2023a; Sun & Yang, 2024; Li et al., 2024).

In summary, existing schemes addressing multi-valued solutions issues either suffer from significant computational expense or lack prediction performance guarantee. In this work, we propose a **generative learning** framework for approximating the solution distribution for non-convex continuous problems with bounded optimality loss and considerable speedup. Our scheme is inspired by the recent success of SDE/ODE-based generative models in complex distribution modeling, such as image generation and combinatorial problems (Ho et al., 2020; Sun & Yang, 2024). However, it is uniquely different in the problem definition, solution optimality, and performance guarantee.

## 3 PROBLEM STATEMENT

We consider the following continuous optimization problem:

$$\min_{x \in \mathcal{K}_c} \quad f(x, c), \tag{1}$$

where $c \in \mathcal{C} \subset \mathbb{R}^d$ denotes the input contextual variables, $x \in \mathcal{K}_c \subset \mathbb{R}^n$ is the decision variables, $\mathcal{K}_c \subset \mathbb{R}^n$ is the compact input-dependent constraint set and $f(x, c) : \mathbb{R}^{n+d} \to \mathbb{R}$ is a continuous objective function (potentially non-convex). Such a formulation in (1) covers wide-ranging engineering applications, including general continuous constrained optimization and relaxed combinatorial problems. The global optimal solutions are denoted as $x_c^* \in X_c^* = \arg\min_{x \in \mathcal{K}_c}\{f(x, c)\}$, where $X_c^*$ is the optimal solution set.

For strictly-convex problems, it has a unique optimal solution $x_c^*$ for each input $c$, corresponding to a single-valued *input-solution mapping* $F : c \to x_c^*$ (Pan et al., 2019; Donti et al., 2021; Amos, 2022). By learning such a mapping and passing new input through the learned mapping to obtain a quality solution instantly, neural networks (NN) have shown remarkable speed improvements in solving convex problems over iterative solvers.

However, non-strictly convex or non-convex problems may exhibit a complexity that allows multiple optimal solutions for identical inputs, resulting in multi-valued input-solution associations. Consequently, the training dataset might include "mixed" data points, reflecting this "one-to-many" mapping. This complexity can impede the trained NN from accurately learning the intended input-solution relationship, resulting in suboptimal solutions. As discussed in Sec. 2, existing studies for continuous constrained optimization endeavor to identify a single possible mapping from the potential multiples, employing various data preparation or processing techniques. Nevertheless, these approaches often compromise between poor optimality performance and high computational complexity.

In contrast, this paper introduces a paradigm shift and leverages NN to learn the mapping from input to solution distribution instead of from input to solution. This method enables us to architect NN schemes that balance low computational complexity with robust optimality performance, thereby more effectively addressing non-convex problems.

### 3.1 INPUT-DEPENDENT SOLUTION DISTRIBUTION AND TRAINING DATA

We make the following assumption on the input-dependent solution distribution to be learned:

**Assumption 1.** The input-dependent solution distribution $p_d(x|c)$ obeys the Boltzmann distribution with inverse temperature parameter as $\beta \geq 0$:

$$p_d(x|c) = \frac{\exp(-\beta f(x, c))}{\int_{\mathcal{K}_c} \exp(-\beta f(x, c))\mathrm{d}x}, \quad \forall x \in \mathcal{K}_c. \tag{2}$$

This assumption is common in ML research and can easily be satisfied (i) when stochastic algorithms are adopted to solve the non-convex problem for general $\beta$, for example, Markov chain Monte Carlo (Chen et al., 2013; Deng et al., 2020) and simulated annealing (Dekkers & Aarts, 1991; Correia et al., 2023), or (ii) when $\beta \to \infty$ and the dataset contains only multiple global optimal solutions, which is aligned with the setting in (Nandwani et al., 2020). This assumption also implies that (i) concentration of probability in high-quality solutions, where $\beta$ can represent the quality of the dataset, and (ii) continuity of the probability density $p_d(x|c)$ on both $x$ and $c$ given the continuous optimization problem in (1), which is critical for the theoretical guarantee for our generative framework.

Given the training dataset sampled from the distribution, in the next section, we will develop a generative learning framework to learn the mapping from input $c$ to solution distribution $p_d(x|c)$. We can then generate quality solutions for new inputs by sampling from the learned distributions.

## 4 OUR GENERATIVE LEARNING FRAMEWORK

To address the challenge of the multi-valued solution mapping problem in Sec. 3, we utilize generative models to learn the input-dependent solution distribution. While there are different generative models to learn a target distribution as discussed in Sec. 2, we select the ODE-based generative model (RectFlow Liu et al. (2022)) because of its strong approximation ability for complex distribution and fast generation process[1].

Our framework, as shown in Fig. 2, consists of three modules designed to (i) model the input-dependent solution distribution for the given "mixed" dataset using the RectFlow model in Sec. 4.1,

---

[1]We also provide a comprehensive comparison of different generative models in the Appendix B.

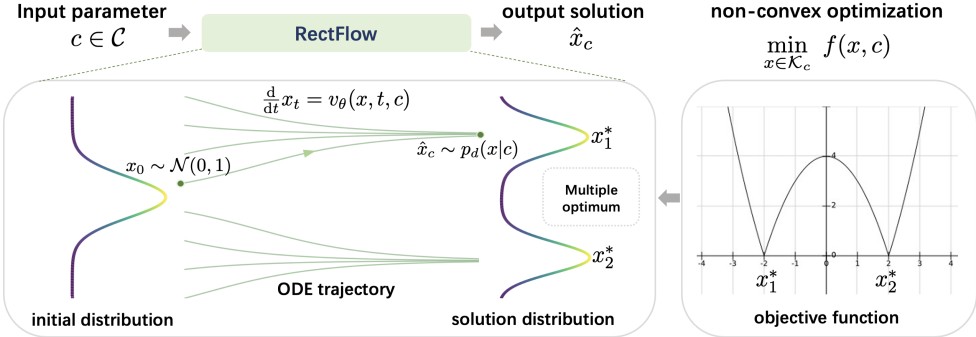

Figure 2: Our proposed Generative Learning framework.

(ii) train NN-approximated vector field for RectFlow in Sec. 4.2, and (iii) sample solution according to the approximated solution distribution for new input parameters in Sec. 4.3, respectively. We characterize the optimality loss and run-time complexity associated with our approach in Sec. 5.

## 4.1 RECTFLOW MODEL FOR INPUT-DEPENDENT SOLUTION DISTRIBUTION

The RectFlow model is a class of ODE-based generative models that employ a vector field to transform a simple initial distribution into a complex target distribution through a forward ODE. Specific to our study in this paper, we select a smooth distribution $q(x)$, e.g., Gaussian, as the initial distribution, and the input-dependent solution distribution $p_d(x|c)$ as the target distribution. Then, the ODE for transformation always exists under Assumption 1 and can be found by solving the following Fokker–Planck (FP) equations (Ambrosio et al., 2005):

$$\partial p(x_t, t)/\partial t = -\nabla \cdot (p(x_t, t)u(x, t, c)), \tag{3}$$
$$dx_t/dt = u(x_t, t, c), \tag{4}$$
$$\text{boundary conditions: } p(x_0, 0) = q(x), \ p(x_1, 1) = p_d(x|c), \tag{5}$$

where $p(x_t, t)$ denotes the PDF of the random variable $x_t$ at $t \in [0, 1]$ and $u(x, t, c) : \mathbb{R}^{n+d+1} \to \mathbb{R}^n$ represent the vector field to be solved. Once $u(x, t, c)$ is obtained, we can generate new solutions obeying the target distribution by sampling from simple prior $x_0 \sim q(x)$ and conducting integration with the target vector field as $x_1 = x_0 + \int_0^1 u(x_t, t, c)dt$; see an illustrative example in Fig. 2.

Although the FP equations seem difficult to solve, existing works have constructed some explicit-form ODE solutions for them shown in (Lipman et al., 2022; Xu et al., 2022; Albergo et al., 2023; Song et al., 2020a; 2023b). We then follow one of the ODE solutions, the RectFlow (Liu et al., 2022), to obtain an explicit expression of a vector field solution, as shown in the following proposition:

**Proposition 4.1.** *The vector field $u(x, t, c)$, denoted as RectFlow, is a solution to the FP equations with boundary conditions $p_0 = q(x)$ and $p_1 = p_d(x|c)$. It is given as:*

$$u(x, t, c) = \mathbb{E}_{x_0 \sim q(x), x_1 \sim p_d(x|c)}[x_1 - x_0 | x_t = x], \tag{6}$$

*where $x_t = (1 - t)x_0 + tx_1$.*

We make the following **remarks**. The vector field solution $u(x, t, c)$ can generate samples following the target distribution through forward ODE integration. However, it can not be directly calculated because of the expectation of the target distribution. On the other hand, given the historical dataset sampled from the target distribution under different input parameters, we can leverage the neural network (NN), denoted as $v_\theta(x, t, c)$, to learn the target vector field in (6), which is continuous over $\mathbb{R}^{n+d} \times [0, 1)$, given the absolutely continuous initial and target distributions Ambrosio et al. (2005). After training, the NN-approximated vector field can be directly used to generate input-dependent solutions by conducting integration.

## 4.2 TRAINING NN-APPROXIMATED VECTOR FIELD

As discussed after Prop. 4.1, the target vector $u(x, t, c)$ in (6) is continuous and can transform a simple distribution to the target solution distribution, but it can not be directly applied. Thus, we

leverage the universal approximation capacity of NN, denoted as $v_\theta(x, t, c)$, to approximate it based on the collected samples from $p_0$ and $p_1$. We employ the following loss function:

$$\mathcal{L}_c(v_\theta) = \int_0^1 \mathbb{E}_{x_0 \sim q(x), x_1 \sim p_d(x|c)}[\|v_\theta(x_t, t, c) - (x_1 - x_0)\|^2]\mathrm{d}t, \tag{7}$$

where $x_t = (1 - t)x_0 + tx_1$, $x_0 \sim q(x)$ is sampled from a fixed prior distribution (e.g., Gaussian), and $x_1 \sim p_d(x|c)$ is the solution sampled from the training dataset. Minimizing the loss function in (7) will guide the NN to learn the target vector field in (6), as shown in the following proposition:

**Proposition 4.2** (Liu et al. (2022)). *The minimizer of the loss function in* (7) *is the target vector field* $u(x, t, c)$ *in* (6), *and a solution to the FP equations with boundary conditions in* (3)-(5).

Therefore, we can employ the loss function in (7) to learn the target vector field solution to the FP equations. Further, to optimize the average performance across different input parameters, we employ the total loss function as follows:

$$\mathcal{L}(v_\theta) = \mathbb{E}_c[\mathcal{L}_c(v_\theta)] = \int_0^1 \mathbb{E}_{x_0 \sim q(x), (c, x_1) \sim p_d(c, x)}[\|v_\theta(x_t, t, c) - (x_1 - x_0)\|^2]\mathrm{d}t. \tag{8}$$

To minimize the loss, we first prepare the training data, including (i) the historical dataset $\{c^i, x_1^i\}_{i=1}^N$ with different input-solution pairs, (ii) samples from fixed Gaussian distribution $x_0 \sim q(x)$, and (iii) uniform sampled time index $t \in [0, 1)$. After that, we can train $v_\theta$ based on the loss function in (8) using optimizers such as Adam (Kingma & Ba, 2014).

### 4.3 Generating solution under new input parameters

After training the NN-approximated vector field $v_\theta$ for the generative model, given new input parameters $c$, we can generate solutions following the target distribution approximately by solving forward ODE as $x_1 = x_0 + \int_0^1 v_\theta(x_t, t, c)\mathrm{d}t$. Such an integration can be efficiently implemented by a numerical solver. For example, the Euler method with a discretized step of $1/k$ is as follows:

$$x_{(i+1)/k} = x_{i/k} + 1/k \cdot v_\theta(x_{i/k}, i/k, c), \quad i = 0, 1, 2, \ldots, k - 1 \tag{9}$$

where $x_0 \sim q(x)$ is sampled from Gaussian distribution. However, owing to the approximation error of the NN vector field and discretization error of numerical ODE solving, the optimality and feasibility of the generated solutions may not be guaranteed. We apply the following strategy to improve the solution quality:

To improve the quality of the predicted solution, we employ a sampling-then-selection strategy. This strategy generates multiple solutions by sampling a batch of initial points $x_0 \sim q(x)$, conducting integration simultaneously to generate candidate solutions, and selecting the one with the minimum objective value and constraint violation.

To ensure the feasibility of generated solutions for continuous constrained optimization problems, we can execute projection operations (Chen et al., 2021a; Liang et al., 2023) to find a nearby feasible point. When solving the relaxed combinatorial problem, after generating a relaxed continuous solution, we can adopt a greedy decoding strategy to recover a feasible discrete solution (Kool et al., 2022; Sun & Yang, 2024).

## 5 Performance Analysis

In this section, we characterize the optimality loss and the run-time complexity for our generative learning, considering factors such as the NN approximation error, ODE discretization step, and the number of sampled solutions. We then discuss the training complexity, scalability, and limitations.

### 5.1 Solution optimality and Run-time complexity

The following theorem characterizes the optimality loss and run-time complexity associated with our generative approach based on the NN-approximated vector field of the ODE-based generative model.

**Theorem 1.** *Recall $u(x, t, c)$ is a solution to the FP equations in (3)-(5). Let $v_\theta$ be the NN-approximate vector field and $\epsilon_\theta$ be the approximation error: $\epsilon_\theta = \mathbb{E}_{x,t}[\|(v_\theta(x, t, c) - u(x, t, c)\|^2]$. Let $\{\hat{x}_{c,i}^k\}_{i=1}^m$ be the generated $m$ solutions using the vector field $v_\theta$ and Euler methods with a stepsize of $1/k$, as detailed in Sec. 4.3. we have*

 • *the probability that the best of the $m$ solutions $\{\hat{x}_{c,i}^k\}_{i=1}^m$ incurs an optimality gap larger than any $\delta > 0$ decreases exponentially to zero as $m$ increases:*

$$\Pr\left(\min_{i=1,...,m}\{f(\hat{x}_{c,i}^k, c)\} \geq f(x_c^*, c) + \delta\right) \leq \left(1 - \Pr(S_c^\delta|c) + C_1\epsilon_\theta^{1/4} + C_2 k^{-1/2}\right)^m, \quad (10)$$

*where $C_1$ and $C_2$ are positive constants depending on Lipschitz conditions of both $u(x, t, c)$ and $v_\theta(x, t, c)$, and $\Pr(S_c^\delta|c) = \int_{S_c^\delta} \exp(-\beta f(x, c)) \mathrm{d}x / \int_{\mathcal{K}_c} \exp(-\beta f(x, c)) \mathrm{d}x$ represents the probability of collected training data lies in the $\delta$-sublevel set $S_c^\delta = \{x \in \mathcal{K}_c \mid f(x, c) \leq f(x_c^*, c) + \delta\}$.*

 • *the run-time complexity of generating a solution $\hat{x}_{c,i}^k$ is $\mathcal{O}(k \cdot l \cdot (n + d)^2)$, where $l$ is the number of layers in the fully connected NN and $n$ and $d$ are dimensions of the decision variable and input parameter of problem in (1), respectively.*

The complete proof is in Appendix A. We make the following **remarks**. First, Theorem 1 elucidates the optimization gap of the solution generated under the following conditions: (i) the quality of training data, represented by $\Pr(S_c^\delta|c)$, (ii) the approximation error of the learned vector field by the NN, denoted as $\epsilon_\theta$, (iii) the number of ODE discretization steps, and (iv) the number of sampled candidate solutions.

To boost the probability of sampling high-quality solutions (i.e., $\delta$ is small), certain strategies can be implemented in the offline training phase. One effective method is to collect a solution dataset of high quality (i.e., $\beta$ is large). This ensures that $p_d(x|c)$ predominantly focuses on the high-quality solutions, and consequently, $\Pr(S_c^\delta|c)$ approaches 1. Furthermore, to reduce $\epsilon_\theta$, the loss function in Eq. (7) needs to be minimized. This can be achieved by utilizing deeper NNs with less model approximation error and incorporating more training data to reduce generalization error. During the online inference phase, the total simulating step $k$ of the ODE solver can be increased to reduce the discretization error. Another strategy is to sample a larger number of candidate solutions and select the most optimal one.

Second, the runtime complexity associated with generating a solution increases linearly with the number of ODE discretization steps and is also influenced by the structure of the adopted NN. Augmenting the number of sampled solutions does not significantly exacerbate the complexity due to the efficiency of batch computation. Therefore, to strike a balance between optimality loss and runtime complexity, it is more advantageous to sample more solutions rather than increasing the number of discretization steps.

## 5.2 TRAINING COMPLEXITY, SCALABILITY, AND LIMITATIONS

To train the NN-approximated vector field within our generative learning framework, we require a training dataset $\mathcal{D}$, which contains different input-solution pairs, samples from a fixed prior distribution (e.g., Gaussian), and uniform time index from a unit interval. Subsequently, at each iteration, we separately sample a batch of $(c, x_1)$, $x_0$, and $t$, and train the NN using the Adam optimizer (Kingma & Ba, 2014). The training computation hinges on the forward-backward propagation of the NN, which can be efficiently executed on a GPU. The principal scalability issues arise during the preparation of the training dataset. In this phase, we need to repeatedly solve the non-convex problem under varied inputs using existing solvers in the offline phase. In the absence of ground-truth solution data, the unsupervised/reinforcement learning approaches for NN training will be explored in future work.

## 6 EXPERIMENTS

We evaluate and compare the performance of our generative learning framework with the RectFlow model and other NN-based schemes for (i) learning a 2-dim complex multi-valued mapping, (ii) solving continuous non-convex problems with multiple solutions, and (iii) solving combinatorial

problems. The NN-based approaches to compare include learning deterministic mappings: (i) **simple NN** to directly predict the solution based on the input parameter; (ii) **hindsight loss** to train multiple NN mappings simultaneously (Li et al., 2018); (iii) **clustering-based** methods to train multiple mappings by partitioning the datasets into clusters of different mappings (Nandwani et al., 2020; Courts & Kvinge, 2021); and learning solution distributions: (iv) **generative adversarial network** (GAN) (Goodfellow et al., 2020); (v) **diffusion-based** model (Song et al., 2020a; Sun & Yang, 2024); (vi) our **generative learning** with RectFlow (Liu et al., 2022). The detailed setting of hyper-parameters and implementation of different approaches are in Appendix C.1.

## 6.1 LEARNING MULTI-VALUED INPUT-SOLUTION MAPPING

To elucidate and juxtapose the various NN-based schemes, we commence with a two-dimensional, albeit complex, example. Consider the setting where a non-convex problem admits six optimal solutions with the same optimal objective value for each input, leading to a 6-valued input-solution mapping as $F(c) = \{3c^2 - 0.5, \ -3c^2 + 0.5, \ c + 1, \ -c + 1, \ c - 1, \ -c - 1\}$.

We then generate a dataset, denoted as $\mathcal{D} = \{c_i, x_i\}_{i=1}^N$, by sampling input-solution pairs from $F(c)$. The constructed input-dependent solution distribution is notably discrete, with support that encompasses no more than six distinct points, implied by the limit $\beta \to \infty$ in Assumption 1. After training, we sample testing input parameters $c$ and feed them into the trained models. The output solutions are visualized in Fig. 3, providing an illustrative comparison of the performance of different models. Additional graphical representations, such as the visualizations of the learned ODE flow, can be found in Appendix C.2. First, the RectFlow model exhibits superior performance in approximating the target multi-valued mappings. In contrast, the simple NN fits an average curve across multiple mappings. The performance of both the hindsight loss-based and cluster-based methods is notably sensitive to the predefined number of potential mappings. The GAN fails to fit such a multi-modal distribution. The diffusion models display an approximation performance similar to that of RectFlow. However, the inherent stochasticity of the backward diffusion process may cause deviations in the generated solution, as indicated by the color dispersion of similar generated solutions.

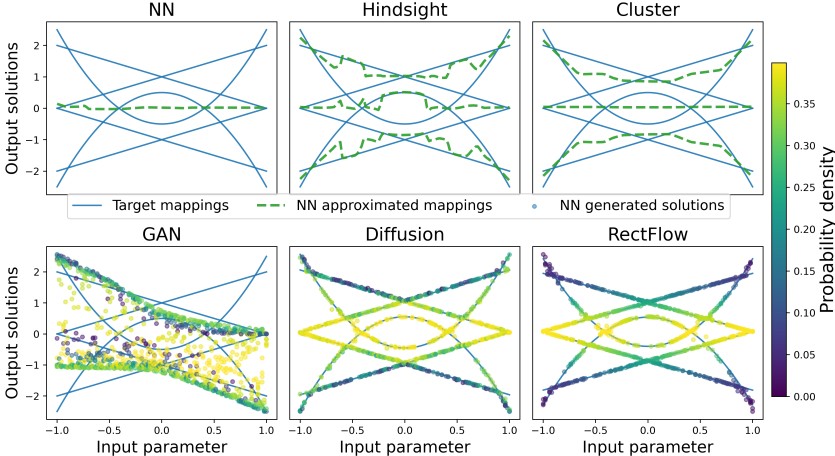

Figure 3: Performance of different models learning multi-valued mapping. The approaches in the first row (NN, Hindsight, and Cluster) aim to learn multiple single-valued mappings, while the ones in the second row (GAN, Diffusion, and RectFlow) learn the solution distribution instead (the color represents the sampling probability).

## 6.2 LEARNING SOLUTION DISTRIBUTION FOR CONTINUOUS NON-CONVEX PROBLEMS

We then test the generative learning with two real-world non-convex problems, including the **inverse kinematics problem** with non-convex constraints and the **wireless power control** with non-convex objectives, where the detailed formulations and data generation settings are in Appendix C.3. These problems are challenging to be solved in real time due to their non-convex constraints and objectives.

As shown in Table 2, first, those approaches aiming to learn deterministic solution mappings fail due to the existence of multiple optimal solutions. When implementing hindsight loss-based or clustering-based methodologies, a precise number of mappings must be designated, which is often

Table 2: Performance comparison on the different continuous non-convex problems

| Problem Metric | Inverse kinematics problem | | | | Wireless power control | | | |
|---|---|---|---|---|---|---|---|---|
| | Error ↓ | Speedup ↑ | Error ↓ | Speedup ↑ | Opt. gap ↓ | Speedup ↑ | Opt. gap ↓ | Speedup ↑ |
| Prob. size | $d=2, n=4$ | | $d=2, n=7$ | | $d=20^2, n=20$ | | $d=40^2, n=40$ | |
| NN | 2.82 | 850 | 4.80 | 1327 | 25.42% | 16 | 30.64% | 49 |
| Hindsight | 1.06 | 479 | 3.11 | 751 | 21.71% | 10 | 28.67% | 28 |
| Cluster | 1.22 | 481 | 3.69 | 757 | 21.49% | 10 | 27.15% | 28 |
| GAN | 0.23 | 435 | 1.53 | 593 | 19.31% | 8 | 36.39% | 16 |
| Diffusion | 0.05 | 226 | 0.35 | 280 | 12.08% | 4 | 13.77% | 6 |
| **RectFlow** | 0.04 | 258 | 0.37 | 348 | 2.94% | 5 | 3.96% | 7 |

[1] $d$ is the input parameter dimension and $n$ is the decision variable dimension.
[2] The error for the inverse kinematics problem represents the constraint violation.
[3] The predicted solutions for the wireless power problem have been projected to its constraint set.

impractical and results in considerable optimality loss. On the other hand, facing such a complex solution distribution, the GAN struggles to capture it and has a large optimality gap. Both the RectFlow and the diffusion-based model can approximate complex solution distributions. However, the adopted RectFlow model outperforms the diffusion-based model by demonstrating a smaller optimality gap and a superior speedup. In essence, the RectFlow model balances optimality and speedup when learning complex, input-dependent solution distributions, making it an excellent choice for tackling the multi-valued input-solution mapping for non-convex problems.

We also conduct an ablation study on the trade-off between optimality and run-time complexity under various parameter settings, such as the number of sampled solutions and discretization steps, in Appendix C.5. This visualization aids in understanding the analysis presented in Theorem 1 and offers further guidance for empirically tuning these parameters.

### 6.3 LEARNING SOLUTION DISTRIBUTION FOR COMBINATORIAL OPTIMIZATION

Table 3: Performance comparison on different non-convex combinatorial problems.

| Problem Metric | Maximum clique | | Maximum independent set | | Maximum cut | |
|---|---|---|---|---|---|---|
| | Opt. gap ↓ | Speedup ↑ | Opt. gap ↓ | Speedup ↑ | Opt. gap ↓ | Speedup ↑ |
| Graph size | 100-node graph: $d=100^2, \ n=100$ | | | | | |
| NN | 22.77% | 64.17 | 23.18% | 78.46 | 12.06% | 330.06 |
| Hindsight | 17.40% | 31.00 | 18.84% | 34.46 | 12.07% | 106.92 |
| Cluster | 23.49% | 33.45 | 19.20% | 30.44 | 99.77% | 139.44 |
| GAN | 20.98% | 36.79 | 16.43% | 29.68 | 100% | 170.78 |
| Diffusion | 5.73% | 14.66 | 5.40% | 14.30 | 8.21% | 71.10 |
| **RectFlow** | 4.87% | 18.06 | 5.59% | 17.02 | 1.35% | 92.69 |

[1] Note that both the optimality gap and speedup presented are based on solutions after greedy decoding. As such, the feasibility is guaranteed to be 100% for all the schemes.

We further generalize our framework to cope with the combinatorial problems. We select three classic cases for this purpose: **the maximum clique**, **the maximum independent set**, and **the maximum-cut problem**. The detailed formulations and data generation settings are in Appendix C.4. As with non-convex continuous problems, combinatorial graph problems display many symmetric solutions. This vast solution space poses significant challenges for established methods that aim to learn single/multiple deterministic solution mappings, as illustrated in Table 3. However, approaches focusing on learning the solution distribution have demonstrated superior empirical performance. Specifically, RectFlow has shown a better balance between optimality and computational efficiency.

## 7 CONCLUSION

We develop a generative learning approach for managing multi-valued input-solution mappings in continuous non-convex problems. It is built upon the ODE-based generative model and learns the mapping from input to solution distribution, a departure from the conventional focus on input-solution mapping. This model, upon receiving new input, samples high-quality solutions from the learned distribution. The optimality loss and runtime complexity of our method are characterized, showing its efficiency. Simulations show that our approach achieves significantly better solution optimality compared to recent NN schemes, while maintaining comparable speedup. This highlights the potential of our method in handling complex multi-valued input-solution mappings, paving the way for future research in machine learning for constrained optimization problems.

## 8 ACKNOWLEDGEMENT

This work is supported in part by a General Research Fund from Research Grants Council, Hong Kong (Project No. 11203122), an InnoHK initiative, The Government of the HKSAR, Laboratory for AI-Powered Financial Technologies, and a Shenzhen-Hong Kong-Macau Science & Technology Project (Category C, Project No. SGDX20220530111203026). The authors would also like to thank the anonymous reviewers for their helpful comments.

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

# A  PROOF FOR THEOREM 1

**Theorem 1**: Recall $u(x, t, c)$ is a solution to the FP equations in (3)-(5). Let $v_\theta$ be the NN-approximate vector field and $\epsilon_\theta$ be the approximation error: $\epsilon_\theta = \mathbb{E}_{x,t}[\|(v_\theta(x, t, c) - u(x, t, c)\|^2]$. Let $\{\hat{x}_{c,i}^k\}_{i=1}^m$ be the generated $m$ solutions using the vector field $v_\theta$ and Euler methods with a stepsize of $1/k$, as detailed in Sec. 4.3. we have

- the probability that the best of the $m$ solutions $\{\hat{x}_{c,i}^k\}_{i=1}^m$ incurs an optimality gap larger than any $\delta > 0$ decreases exponentially to zero as $m$ increases:

$$\Pr\left(\min_{i=1,\dots,m}\{f(\hat{x}_{c,i}^k, c)\} \geq f(x_c^*, c) + \delta\right) \leq \left(1 - \Pr(S_c^\delta|c) + C_1\epsilon_\theta^{1/4} + C_2 k^{-1/2}\right)^m, \quad (11)$$

where $C_1$ and $C_2$ are positive constants depending on Lipschitz conditions of both $u(x, t, c)$ and $v_\theta(x, t, c)$, and $\Pr(S_c^\delta|c) = \int_{S_c^\delta}\exp(-\beta f(x, c))\mathrm{d}x / \int_{\mathcal{K}_c}\exp(-\beta f(x, c))\mathrm{d}x$ represents the probability of collected training data lies in the $\delta$-sublevel set $S_c^\delta = \{x \in \mathcal{K}_c \mid f(x, c) \leq f(x_c^*, c) + \delta\}$.

- the run-time complexity of generating a solution $\hat{x}_{c,i}^k$ is $\mathcal{O}(k \cdot l \cdot (n + d)^2)$, where $l$ is the number of layers in the fully connected NN and $n$ and $d$ are dimensions of the decision variable and input parameter of problem in (1), respectively.

*Proof.* **(a) Optimality loss of generated solutions**

We first investigate the probability of collected training data lies in the $\delta$-sublevel set $S_c^\delta = \{x \in \mathcal{K}_c \mid f(x, c) \leq f(x_c^*, c) + \delta\} \subseteq \mathcal{K}_c$.

$$\Pr(S_c^\delta|c) = \Pr(f(x_c, c) \leq f(x_c^*, c) + \delta\}) \quad (12)$$

$$= \int_{\mathcal{K}_c} p_d(x|c)I(f(x_c, c) \leq f(x_c^*, c) + \delta)\mathrm{d}x \quad (13)$$

$$= \frac{\int_{S_c^\delta}\exp(-\beta f(x, c))\mathrm{d}x}{\int_{\mathcal{K}_c}\exp(-\beta f(x, c))\mathrm{d}x}, \quad (14)$$

when $\beta = 0$, the solution distribution is correspondingly a uniform distribution, and the probability is the ratio of two volumes of sets as $\Pr(S_c^\delta|c) = \mathrm{Vol}(S_c^\delta) / \mathrm{Vol}(\mathcal{K}_c) \leq 1$. With the increase of $\beta$, the solution distribution concentrates on the solutions with better objective values. When $\beta \to \infty$, the probability converges to $\Pr(S_c^\delta|c) \to 1$, such that almost all sampled solutions are within $S_c^\delta$.

The next step is to consider the probability of sampling from the $\delta$ sub-level set with an approximated generative model, denoted as $\Pr(S_c^\delta|c, \theta)$. To proceed, let $u(x, t, c)$ and $p_t(x)$ be a target solution (e.g., RectFlow) of the Fokker–Planck equation with desired boundary conditions $p_0 = q(x)$ and $p_1 = p_d(x|c)$ under input parameter $c$. $v_\theta(x, t, c)$ is the NN-approximated vector field, and $q_t(x)$ is the corresponding PDF driven by $v_\theta(x, t, c)$ under input parameter $c$. $\hat{q}_t(x)$ is the PDF driven by $v_\theta(x, t, c)$ under Euler methods of $k$ discretization steps. The initial distribution is the same as $p_0(x) = q_0(x) = \hat{q}_0(x) = \mathcal{N}(0, I_n)$. The terminal distributions are denoted as $x_c \sim p_1(x) = p_d(x|c)$, $\hat{x}_c \sim q_1(x) = p_\theta(x|c)$, and $\hat{x}_c^k \sim \hat{q}_1(x) = p_\theta^k(x|c)$ respectively.

Then we can can bound $\Pr(S_c^\delta|c, \theta)$ as:

$$\Pr(S_c^\delta|c, \theta) = \Pr(f(\hat{x}_c^k, c) \leq f(x_c^*, c) + \delta) \quad (15)$$

$$= \Pr(S_c^\delta|c, \theta) - \Pr(S_c^\delta|c) + \Pr(S_c^\delta|c) \quad (16)$$

$$\geq \Pr(S_c^\delta|c) - |\Pr(S_c^\delta|c, \theta) - \Pr(S_c^\delta|c)| \quad (17)$$

$$\geq \Pr(S_c^\delta|c) - \sup_{A \subset \mathcal{K}_c}\{|\Pr(A|c, \theta) - \Pr(A|c)|\} \quad (18)$$

$$= \Pr(S_c^\delta|c) - \mathrm{TV}(p_\theta^k(x|c), p_d(x|c)), \quad (19)$$

where the total variation distance between two distributions $p(x)$ and $q(x)$ with compact support set of $X$ is defined as $\mathrm{TV}(p, q) = \sup_{A \subset X}\{|p(A) - q(A)|\} \in [0, 1]$.

Considering the approximation error of the generative model on the total variation distance, the probability $\Pr(S_c^\delta|c, \theta)$ may be low. Therefore, we propose the sampling-then-selection strategy to

improve the optimality. Suppose we select the best solution from $\{\hat{x}_{c,i}^k\} \sim p_\theta^k(x|c)$. The probability of at least sampling one solution within the $\delta$ sub-level set is bounded as:

$$\Pr(\min_{i=1,\dots,m} \{f(\hat{x}_{c,i}, c)\} \geq f(x_c^*, c) + \delta\}) \tag{20}$$

$$= (1 - p_\theta(S_c^\delta|c))^m \tag{21}$$

$$\leq (1 - \Pr(S_c^\delta|c) + \mathrm{TV}(p_\theta^k(x|c), p_d(x|c)))^m, \tag{22}$$

Therefore, to maximize the probability of sampling high-quality solution, we can (i) prepare a high-quality solution dataset where $\Pr(S_c^\delta|c)$ is close to 1, (ii) minimize the total variation distance $\mathrm{TV}(p_\theta^k(x|c), p_d(x|c))$ between the approximated solution distribution via the generative model and the data distribution and (iii) increase the number of generated solutions in the online inference phase.

We then bound the total variation distance $\mathrm{TV}(p_\theta^k(x|c), p_d(x|c))$ considering the approximation error of NN vector field and discretization step. We split the total variation distance into two parts:

$$\mathrm{TV}(p_d(x|c), p_\theta^k(x|c)) \leq \underbrace{\mathrm{TV}(p_d(x|c), p_\theta(x|c))}_{\text{learning error}} + \underbrace{\mathrm{TV}(p_\theta(x|c), p_\theta^k(x|c))}_{\text{discretization error}} \tag{23}$$

where the learning error evaluates the distance between data distribution and generated distribution by exact integration, and the discretization error evaluates the distance between generated distributions under exact integration and Euler methods of $k$ discretization steps.

To tackle the learning error, according to the Pinsker's inequality, we have

$$\mathrm{TV}(p_d(x|c), p_\theta(x|c)) \leq \sqrt{1/2 \, \mathrm{KL}(p_d(x|c) \mid p_\theta(x|c))}, \tag{24}$$

The KL divergence between two distributions $p(x)$ and $q(x)$ with compact support set of $X$ is defined as $\mathrm{KL}(p \mid q) = \int_{x \in X} p(x) \log \frac{p(x)}{q(x)} \mathrm{d}x$.

According to Theorem 3.1 in (Lu et al., 2022), the KL divergence can be bounded as:

$$\mathrm{KL}(p_d(x|c) \mid p_\theta(x|c)) \leq \int_0^1 \int_x p_t(x)(u(x,t,c) - v_\theta(x,t,c))^\top (\nabla_x \log p_t(x) - \nabla_x \log q_t(x)) \mathrm{d}x \mathrm{d}t \tag{25}$$

According to the Cauchy–Schwarz inequality, we can decompose the bound in Eq. (25) as:

$$\mathrm{KL}(p_d(x|c) \mid p_\theta(x|c)) \leq \underbrace{\mathbb{E}_{t,x}[\|v_\theta(x,t,c) - u(x,t,c)\|^2]^{1/2}}_{\text{NN approximation error as } \epsilon_\theta} \underbrace{\mathbb{E}_{t,x}[\|\nabla_x \log p_t(x) - \nabla_x \log q_t(x)\|^2]^{1/2}}_{\text{Fisher divergence as } D_F(p_t \mid q_t)}, \tag{26}$$

where the first part represents the NN approximation error as $\epsilon_\theta = \mathbb{E}_{t,x}[\|(v_\theta(x,t,c) - u(x,t,c)\|^2]$, and the second part denotes the Fisher divergence as $D_F(p_t \mid q_t) = \mathbb{E}_{t,x}[\|\nabla_x \log p_t(x) - \nabla_x \log q_t(x)\|^2]$.

We make the following remakes for the upper bound of KL divergence in (26). Firstly, this upper bound represents the product of the Neural Network (NN) approximation error in learning the target vector field and the Fisher divergence between two probability flows. The goal of our loss function (as shown in Eq. (7)) is to minimize the approximation error. As for the Fisher divergence, it can be polynomially bounded under certain Lipschitz conditions of the target vector field and the NN-approximated one, as stipulated by Theorem 4.7 in (Chen et al., 2023). Moreover, as demonstrated by Theorem 3.2 in (Lu et al., 2022), Fisher divergence can also be optimized to expedite the convergence of the KL distance. This optimization can be achieved by incorporating certain high-order regularization terms into the loss function. However, due to computational constraints, we concentrate on minimizing $\epsilon_\theta$ as outlined in the loss function (refer to Eq. 7).

Next, by substituting the KL divergence term in Pinsker's inequality (as given in Eq. (24)) with the upper bound provided in Eq. (26), we can bound the first part of the total variation distance as follows:

$$\mathrm{TV}(p_d(x|c), p_\theta(x|c)) \leq \frac{\sqrt{2}}{2} (D_F(p_t \mid q_t)\epsilon_\theta)^{1/4} \tag{27}$$

We then tackle the discretization error under Euler methods with $k$ discretization steps.

$$x_{(i+1)/k} = x_{i/k} + \frac{1}{k} v_\theta(x_{i/k}, i/k, c), \quad i = 0, 1, 2, \cdots, k-1 \tag{28}$$

where the initial points $x_0 \sim q(x)$ are sampled from Gaussian distribution.

Let the Lipschitz of the NN-approximated vector field over $x$ and $t$ and $L_{v_\theta}^x$ and $L_{v_\theta}^t$, respectively. Suppose the stepsize $1/k$ is small enough (e.g., $k \geq 2L_{v_\theta}^x$ as in Lemma C.3 in (Chen et al., 2023)) that the discretized ODE trajectory is invertible.

We denote the valid vector field under Euler discretization as $\hat{v}_\theta$, where $\hat{v}_\theta(x, t, c) = v_\theta(x, t, c)$ for $t \in \{0, \frac{1}{k}, ..., \frac{i}{k}, ..., \frac{k-1}{k}\}$. The distance between the continuous vector field and the discretized vector field can be bounded as:

$$\|v_\theta(x, t, c) - \hat{v}_\theta(x, t, c)\| \tag{29}$$
$$= \|v_\theta(x, t, c) - v_\theta(x, \lfloor tk \rfloor / k, c) + v_\theta(x, \lfloor tk \rfloor / k, c) - \hat{v}_\theta(x, t, c)\| \tag{30}$$
$$\leq \|v_\theta(x, t, c) - v_\theta(x, \lfloor tk \rfloor / k, c)\| + \|v_\theta(x, \lfloor tk \rfloor / k, c) - \hat{v}_\theta(x, t, c)\|, \tag{31}$$

where we introduce an auxiliary term, $v_\theta(x, \lfloor tk \rfloor / k, c)$, which represents the vector field at the previous discretized time point.

Next, we bound the two terms in Eq. 31, respectively. For the first term, consider the Lipschitz condition of the NN-approximated vector field with respect to $t$ and derive an upper bound as $\|v_\theta(x, t, c) - v_\theta(x, \lfloor tk \rfloor / k, c)\| \leq L_{v_\theta}^t \frac{1}{k}$.

For the second term, we consider the equivalence between the continuous and the discretized vector fields on the discretized points. Specifically, for a stimulating trajectory $\hat{x}_t$ with $\hat{v}_\theta$, $\hat{v}_\theta(\hat{x}_t, t, c) = \hat{v}_\theta(\hat{x}_{t+h}, t+h, c)$ for $t \in \{0, \frac{1}{k}, ..., \frac{i}{k}, ..., \frac{k-1}{k}\}$ and $h \in [0, 1/k]$. Therefore, we have $\hat{v}_\theta(x, t, c) = \hat{v}_\theta(x - \hat{v}_\theta(x, t, c)(t - \lfloor tk \rfloor / k), \lfloor tk \rfloor / k, c)$. We then consider the Lipschitz condition of the NN-approximated vector field with respect to $x$ and derive an upper bound as $\|v_\theta(x, \lfloor tk \rfloor / k, c) - \hat{v}_\theta(x, t, c)\| = \|v_\theta(x, \lfloor tk \rfloor / k, c) - v_\theta(x - \hat{v}_\theta(x, t, c)(t - \lfloor tk \rfloor / k), \lfloor tk \rfloor / k, c)\| \leq L_{v_\theta}^x \hat{v}_\theta(x, t, c) \frac{1}{k}$.

Combining the upper bounds of the two terms together, we have:

$$\|v_\theta(x, t, c) - \hat{v}_\theta(x, t, c)\| \leq L_{v_\theta}^t \frac{1}{k} + L_{v_\theta}^x \hat{v}_\theta(x, t, c) \frac{1}{k} \tag{32}$$

Under Euler methods with $k$ discretization steps, the KL divergence is similarly bounded as

$$\text{KL}(p_\theta(x|c) \mid p_\theta^k(x|c)) \leq \mathbb{E}_{t,x}[\|v_\theta(x, t, c) - \hat{v}_\theta(x, t, c)\|^2]^{1/2} D_F(q_t \mid \hat{q}_t)^{1/2}, \tag{33}$$

For the distance between vector fields, we can bound it under the Lipschitz condition according to Eq. (32):

$$\mathbb{E}_{t,x}[\|v_\theta(x, t, c) - \hat{v}_\theta(x, t, c)\|^2] \leq (1/k)^2 \mathbb{E}_{t,x}[(L_{v_\theta}^t + L_{v_\theta}^x \hat{v}_\theta(x, t, c))^2] \tag{34}$$

According to Pinsker's inequality (as given in Eq. (24)), the second part of the total variation distance can be bounded as:

$$\text{TV}(p_\theta(x|c), p_\theta^k(x|c)) \leq \frac{\sqrt{2}}{2} (D_F(q_t \mid \hat{q}_t) \mathbb{E}_{t,x}[(L_{v_\theta}^t + L_{v_\theta}^x \hat{v}_\theta(x, t, c))^2])^{1/4} (1/k)^{1/2} \tag{35}$$

Therefore, combining the two upper bounds in Eq. (27) and Eq. (35) together, the total variation distance between the target distribution and the one under NN approximation error and ODE discretization error can be bounded as

$$\text{TV}(p_d(x|c), p_\theta^k(x|c)) \leq \frac{\sqrt{2}}{2} (D_F(p_t \mid q_t) \epsilon_\theta)^{1/4} \tag{36}$$

$$+ \frac{\sqrt{2}}{2} (D_F(q_t \mid \hat{q}_t) \mathbb{E}_{t,x}[(L_{v_\theta}^t + L_{v_\theta}^x \hat{v}_\theta(x, t, c))^2])^{1/4} (1/k)^{1/2} \tag{37}$$

Let the constant terms as $C_1 = \frac{\sqrt{2}}{2} (D_F(p_t \mid q_t))^{1/4}$ and $C_2 = \frac{\sqrt{2}}{2} (D_F(q_t \mid \hat{q}_t) \mathbb{E}_{t,x}[(L_{v_\theta}^t + L_{v_\theta}^x \hat{v}_\theta(x, t, c))^2])^{1/4}$.

By substituting the TV distance in Eq. (22) with the bound above, we have the final result:

$$\Pr(\min_{i=1,...,m}\{f(\hat{x}_{c,i}, c)\} \geq f(x_c^*, c) + \delta\}) \leq (1 - \Pr(S_c^\delta|c) + \mathrm{TV}(p_\theta^k(x|c), p_d(x|c)))^m \quad (38)$$

$$\leq \left(1 - \Pr(S_c^\delta|c) + C_1\epsilon_\theta^{1/4} + C_2k^{-1/2}\right)^m \quad (39)$$

**(b) Run-time complexity of generation process**

The run-time complexity involves the forward calculation of the neural network, the number of discretized steps $k$, and the number of samples $m$.

Considering a fully connected NN represented vector field $v_\theta(x, t, c)$ with input dimension of $n+d+1$, output dimension of $n$, $l$ layers, and $\mathcal{O}(n+d)$ neurons on each hidden layer. The run-time complexity for this NN is as $M = \mathcal{O}(l \cdot (n+d)^2)$. On the other hand, we adopt the transformer encoder for the graph problem. Suppose a $n$-node graph with node embedding of dimension $d$. The run-time complexity for a $l$ layer transformed encoder is of $M = \mathcal{O}(l \cdot (n^2 \cdot d + n \cdot d^2))$.

The integration of ODE is conducted sequentially, which needs to NN forward calculation at each discretized point. Therefore, the total run-time complexity for generating a single solution is $\mathcal{O}(Mk)$.

$\square$

# B COMPARISON OF GENERATIVE MODELS

We compare the performance of existing generative models for distribution approximation using several metrics. These include their capacity to capture various modes of general distributions, the quality of the samples they generate, and the speed of the sampling process. A summary of this comparison is presented in Table 4.

Table 4: Comparison of different generative models in consideration.

| Generative model | Mode Diversity | High Quality Samples | Fast Sampling |
|---|---|---|---|
| Auto-regressive model | ✗ | ✓ | ✗ |
| Normalizing flow | ✓ | ✗ | ✓ |
| Variational auto-encoder | ✓ | ✗ | ✓ |
| Generative adversarial network | ✗ | ✓ | ✓ |
| SDE-based (Diffusion model) | ✓ | ✓ | ✗ |
| ODE-based (RectFlow) | ✓ | ✓ | ✓ |

# C EXPERIMENT SETTINGS

## C.1 COMPARED APPROACHES

We adopt the following approaches in our experiments, including NN-based and generative model based:

• **NN**: a neural network $F : \mathbb{R}^d \to \mathbb{R}^n$ is trained by directly minimizing the prediction mean square error as:

$$\min_F \mathbb{E}_{(c,x)\sim p_d(x,c)}[\|F(c) - x\|^2]. \quad (40)$$

After training, the solution is directly output by the trained model $F(c)$ given new input parameters.

• **Hindsight loss**: multiple neural networks $\{F_i\}_{i=1}^h$ are trained simultaneously based on the hindsight loss (Li et al., 2018):

$$\min_{F_i} \mathbb{E}_{(c,x)\sim p_d(x,c)}[\min_{i=1,...,h}\{\|F_i(c) - x\|^2\}]. \quad (41)$$

After training, multiple solutions are predicted by each model $F_i(c)$ for $i = 1, ..., h$, and we select the best one.

• **Clustering-based approach**: multiple predictors $\{F_i\}_{i=1}^h$ can also be trained by adding an additional NN-based clustering/selection model: $W_i : \mathbb{R}^{d+n} \to \{0, 1\}$, The training loss is as:

$$\min_{F_i, W_i} \mathbb{E}_{(c,x) \sim p_d(x,c)} [\| \sum_{i=1}^h W_i(c, x) F_i(c) - x \|^2] \tag{42}$$

$$\text{s.t.} \sum_{i=1}^h W_i(c, x) = 1, \quad W_i(c, x) \in \{0, 1\}, \tag{43}$$

where the constraint $\sum_{i=1}^h W_i(c, x) = 1$ is to assign one cluster for an input-solution pair. We adopt the Gumbel Softmax layer to reparameterize the constraint and optimize the loss function jointly (Jang et al., 2016). After training, we predict multiple solutions and select the best one.

• **GAN**: The generator $G : \mathbb{R}^{d+n} \to \mathbb{R}^n$ is to transform a simple prior distribution to a complex target distribution, and the discriminator $G : \mathbb{R}^{d+n} \to (0, 1)$ is to judge the quality of generated samples. The loss is defined as:

$$\min_G \max_D \mathbb{E}_{(c,x) \sim p_d(x,c)}[\log D(x, c)] + \mathbb{E}_{c, z \sim \mathcal{N}(0, I_n)}[\log(1 - D(G(z, c)))], \tag{44}$$

After training, we sample from the latent distribution $z \sim \mathcal{N}(0, I_n)$, feed them into the generator as $x = G(z, c)$, and select the best solution.

• **Diffusion**: we adopt the denoising diffusion implicit model (DDIM) with default parameter settings in (Song et al., 2020a; Sun & Yang, 2024).

• **RectFlow**: we adopted the rectified flow model as introduced in Sec. 4.

For the **Hindsight** and **Cluster** approaches, we have chosen four predictors, which are implemented via a multi-head prediction mechanism in the final layer, grounded on common embedding layers. For those generative approaches, including **GAN**, **Diffusion**, and **RectFlow**, we have employed the sampling-then-selection strategy to enhance solution quality. Specifically, we sample 1,000 initial points and generate candidate solutions simultaneously in a batch. For the **Diffusion** and **RectFlow** model, we have set the discretization steps to 10.

## C.2 EXPERIMENT SETTINGS FOR TOY EXAMPLE

We consider the following multi-valued mapping:

$$F(c) = \{3c^2 - 0.5, \ -3c^2 + 0.5, \ c + 1, \ -c + 1, \ c - 1, \ -c - 1\} \tag{45}$$

We then sample a dataset, $\mathcal{D} = \{c_i, x_i\}_{i=1}^N$, using $F(c)$, where the input parameter $c$ is uniformly sampled from $[-1, 1]$ and the solution is randomly returned by one of the six mappings in $F(c)$. After training, we sample input parameters $c$ from the testing dataset, feed them into the trained models, and visualize generated solutions.

For the RectFlow model, the NN-approximated vector field under different input parameters is shown in Fig. 4.

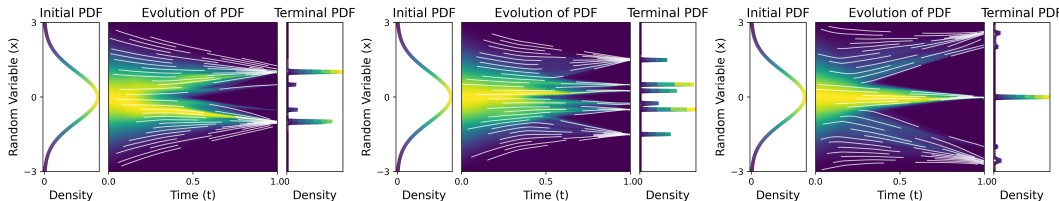

Figure 4: Probability flow of NN approximated RectFlow under $c = 0$ (left), $c = 0.5$ (center), and $c = 1$ (right), respectively.

### C.3 EXPERIMENT SETTINGS FOR CONTINUOUS NON-CONVEX PROBLEMS

We focus on two challenging, continuous non-convex optimization problems:

(i) **Inverse Kinematics Problem:** in real-time control of robotic arms, given a target position, the system must solve for the robotic configuration, which includes determining the angles for each joint. This process involves solving a system of non-linear equations, a task that is notoriously complex due to its non-convex nature. The inverse kinematics problem is a fundamental challenge in robotic control, primarily because a single target position can correspond to multiple feasible configurations. This multiplicity adds complexity to existing NN-based methods.

Specifically, consider a robot arm with $K$ degrees of freedom (DOF), where the length of arms is denoted as $[L_1, \cdots, L_K]$, given a target position $(x, y) \in \mathbb{R}^2$, the optimal angles $\alpha \in \mathbb{R}^K$ for joints is solved by the following non-convex optimization problems:

$$\min_{\alpha \in \mathbb{R}^K} \quad f(\alpha) \tag{46}$$

$$\text{s.t.} \quad \sum_{i=1}^{k} L_i \cos(\sum_{j=1}^{i} \alpha_j) = x \tag{47}$$

$$\sum_{i=1}^{k} L_i \sin(\sum_{j=1}^{i} \alpha_j) = y \tag{48}$$

$$\alpha_{\min} \leq \alpha \leq \alpha_{\max} \tag{49}$$

(ii) **Wireless Power Control:** in wireless networks, power control is a critical issue. The task is to optimize the transmitter power levels in order to maximize network capacity while minimizing energy consumption. The problem is non-convex due to the interaction between the transmitted power and the resulting interference levels. It also admits multiple optimal solutions given the same input, making it a challenging task for existing NN-based methods.

Specifically, consider a network consisting of $K$ single-antenna transceiver pairs. Denote the $K \times K$ channel condition matrix by $h$, where $h_{ii}$ is the direct channel between the $i$-th transmitter and $i$-th receiver, and $h_{ij}$ is the interference channel from the $i$-th transmitter to the $j$-th receiver. Let $\sigma_i$ denote the noise power at the $i$-th receiver. The optimal power level $p \in \mathbb{R}^K$ is solved by the following non-convex optimization:

$$\max_{p \in \mathbb{R}^K} \quad \sum_{i=1}^{K} \alpha_i \log \left( 1 + \frac{|h_{ii}|^2 p_i}{\sum_{j \neq i} |h_{ij}|^2 p_j + \sigma_i^2} \right) \tag{50}$$

$$\text{s.t.} \quad p^{\min} \leq p \leq p^{\max} \tag{51}$$

In both cases, our goal is to leverage the generative learning framework to tackle these complex, non-convex optimization problems that require real-time solutions and can have multiple optimal solutions for the same input. We use similar experimental settings as outlined in (Ardizzone et al., 2018) and (Sun et al., 2017) to gather datasets for the inverse kinematics problem and wireless power control, respectively. We collected 10,000 training instances and conducted tests on an additional 1,024 instances.

The performance metrics we use include the error for the inverse kinematics problems, which evaluates the Euclidean distance between the target position and the position achieved using the predicted configuration. For the wireless power control problem, we measure the optimality gap, comparing it with the target objective value ascertained by the iterative algorithm. For assessing speedup, we chose different iterative algorithms as baselines. Specifically, we used the cross-entropy methods (De Boer et al., 2005) as the iterative solver to derive the optimal solutions for the inverse kinematics problem. For the wireless power control problem, we employed the block gradient descent (Sun et al., 2017) as the iterative solver to obtain (locally) optimal solutions.

## C.4 EXPERIMENT SETTINGS FOR COMBINATORIAL PROBLEMS

In our study, we address three combinatorial problems over a graph, denoted as $G = (V, E)$, wherein $V$ embodies the set of nodes, and $E$ signifies the set of edges. The associated complementary graph is represented as $\overline{G} = (V, \overline{E})$.

The three combinatorial problems are defined as follows:

$$\text{Maximum clique:} \quad \max_x \sum_{i \in V} x_i \tag{52}$$
$$\text{s.t. } x_i + x_j \leq 1, \ \forall (i, j) \in \overline{E}$$
$$x_i \in \{0, 1\}, \ \forall i \in V$$

$$\text{Maximum independent set:} \quad \max_x \sum_{i \in V} x_i, \tag{53}$$
$$\text{s.t. } x_i + x_j \leq 1, \ \forall (i, j) \in E$$
$$x_i \in \{0, 1\}, \ \forall i \in V$$

$$\text{Maximum cut:} \quad \max_x \sum_{(i,j) \in E} \frac{1 - x_i x_j}{2} , \tag{54}$$
$$\text{s.t. } x_i \in \{-1, 1\}, \ \forall i \in V$$

For data collection, we sampled Erdos–Rényi random graphs (Erdős et al., 1960) as the instance dataset. For graphs with fewer than 100 nodes, we used the Gurobi solver to procure globally optimal solutions. For larger graphs, we relied on NetworkX heuristics to identify locally optimal solutions. Our dataset comprised of 10,000 training instances, and we conducted tests on an additional 1,024 instances. The performance metrics included the optimality gap and speedup relative to an iterative solver.

In terms of Neural Network (NN)-based methods, we employed a Transformer encoder to handle the varied graph inputs (Kool et al., 2018). Given the optimal solution involves binary variables with additional constraints, we used a greedy decoding strategy to convert the generated continuous solutions back into discrete feasible ones (Sun & Yang, 2024).

## C.5 ABLATION STUDY FOR GENERATIVE LEANRING

In this section, we evaluate our generative framework under different parameter settings, specifically varying the number of sampled solutions $m \in \{1, 10, 100, 200, 500, 800, 1000\}$ and discretization steps $k \in \{1, 10, 100, 200, 500, 800, 1000\}$.

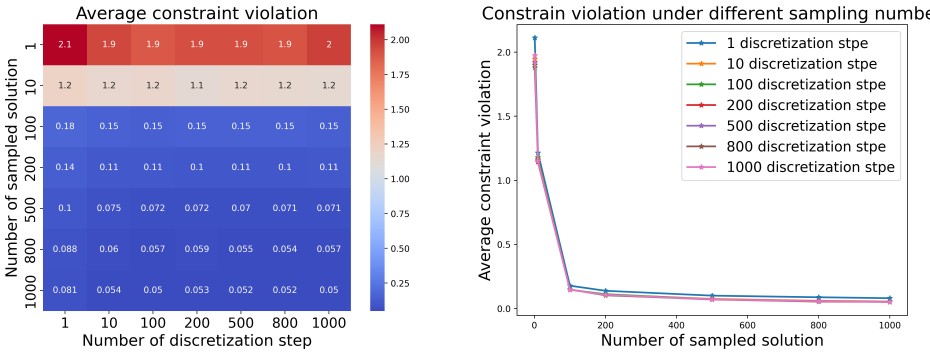

Figure 5: Performance comparison on the solution quality for inverse kinematics problem.

We perform this performance comparison using the inverse kinematics problem ($d = 2, n = 4$). Our evaluation metrics include average constraint violation and average inference time, as depicted in Figures 5 and 6. Our observations are as follows:

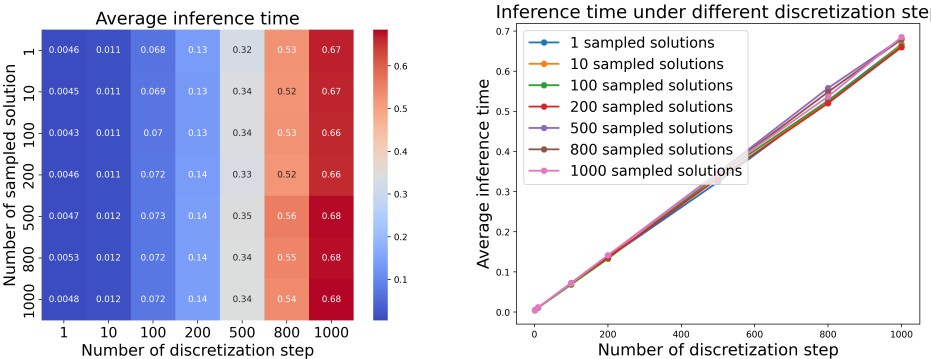

Figure 6: Performance comparison on the inference time for inverse kinematics problem.

First, regarding solution quality, as measured by constraint violation (Figure 5), we notice that the number of sampled solutions significantly impacts solution quality compared to the number of discretization steps. Second, when considering runtime complexity, as gauged by inference time (Figure 6), we find that it scales linearly with the number of discretization steps. However, it is not sensitive to the increasing number of samples. This behavior is due to the use of batch computation.

Consequently, our performance comparison not only supports the assertions made in Theorem 1 but also provides insights for parameter tuning, thereby enabling the generation of better solutions.

