# OpenReview forum: "Generative Learning for Solving Non-Convex Problem with Multi-Valued Input-Solution Mapping"
_ICLR.cc/2024/Conference — ICLR 2024 poster_

### Official Review · Reviewer_85JM · 2023-10-30

**Soundness:** 4 excellent
**Presentation:** 4 excellent
**Contribution:** 4 excellent
**Rating:** 8
**Confidence:** 4

**Summary:**

This paper considers the problem of learning neural approximations of multi-valued input solution mappings arising from solutions of non-convex optimization problems. Previous work on learning single-valued mappings struggled to approximate such multi-valued mappings, compromising between poor optimality performance and high computational complexity. This paper proposes a generative approach based on rectified flows, which is inspired by the recent successes of generative models for modeling complex multi-modal distributions.

The authors propose to process multiple samples in parallel, project them onto the feasible region, and keep only the best one, leading to high quality feasible samples.
This is also quantified in a theorem, which gives an upper bound on the probability of selecting a sample that has optimality gap larger than $\delta$, elucidating the dependence of sample quality on the dataset quality, the approximation error of the learned vector field, the number of ODE discretization steps and the number of samples. Additionally the runtime complexity is characterized, giving insights into the tradeoff between increasing the number of samples and increasing the number of ODE discretization steps.
Experimentally, the proposed approach is compared to various baseline on a synthetic multi-valued dataset and three graph-based combinatorial optimization problems. The proposed approach is shown to strongly outperform previous neural-network based approaches in terms of optimality gap while preserving a strong speedup in runtime.

**Strengths:**

I really enjoyed reading this paper. It is built on a great idea of applying the recently popular diffusion-based models to learning solutions of optimization problems. The method and results are presented with great clarity, and the text reads very well overall. The quality of the paper is high, with interesting theoretical insights that have useful implications, as well as convincing experiments.
I believe this paper makes a very significant contribution, and I agree with the authors that it could pave the way for future research in learning multi-valued solution mappings of optimization problems.

**Weaknesses:**

- Some parts of the proof of the theorem are a bit unclear, see questions.
- The paper could have benefitet from more diversity in the experiments, e.g. including one of the motivating applied examples from the introduction could have made the paper even more of a slam dunk (e.g. AC optimal power flow problems in real-time power grid operations, semi-definite programming-based real-time scheduling, coding operations in modern wireless communication systems). However, I believe the given experimental evidence still makes it a good paper and I believe is sufficient for acceptance.

**Questions:**

- The paragraph after equation 27, in particular how equation 28 follows, was unclear to me.
- In the proof of the theorem, some Lipschitz assumptions are made. It would be good to make these explicit in the theorem statement, in which they are currently not mentioned.
- The step from equation 31 to equation 32 could be explained in a bit more detail.

---

> ### Author Response · Authors · 2023-11-17
> **Response to Comment 1&2 for Reviewer 85JM**
>
> We thank the reviewer for the time and effort in reviewing our paper and the positive comments. We address the concerns one by one in the following.
>
> **Comment 1: Some parts of the proof of the theorem are a bit unclear, see questions.**
>
> **Response:**
> We appreciate your attention to detail in reviewing the proof of our theorem. In response to your feedback, we have revised and clarified those parts of the proof in our revised manuscript.
>
> **Comment 2: The paper could have benefited from more diversity in the experiments, e.g. including one of the motivating applied examples from the introduction could have made the paper even more of a slam dunk.**
>
> **Response:**
> Thanks for your feedback. We agree with your viewpoint that performance evaluation on more diverse problems could further enhance our paper.
>
> Consequently, we have conducted additional experiments for two real-world, non-convex problems featuring more complex problem structures.
> - The first is the **inverse kinematics problem**, which involves determining the optimal robot configuration for a given target position by solving a system of intricate non-linear equations.
> - The second is the **wireless power control problem**, which requires finding the optimal power level for a given transmission configuration by solving a non-convex optimization problem.
>
> Both problems demand real-time solutions, and the existence of multiple solutions presents a challenge to current neural network-based methods. The new experimental results are presented in **Section 6.2 on page 9** of the revised manuscript, and they further illustrate the effectiveness of our proposed method in solving non-convex problems with multi-valued mappings.

---

> ### Author Response · Authors · 2023-11-17
> **Response to Question 1&2&3 for Reviewer 85JM**
>
> **Question 1: The paragraph after equation 27, in particular how equation 28 follows, was unclear to me.**
>
> **Response:**
> Thank you for your feedback. We make the following clarifications:
>
> First, Eq. (27) shows that the KL divergence is bounded by the product of the Neural Network approximation error and the Fisher divergence: ${\rm KL}(p_d(x|c)|p_{\theta}(x_c))\le \sqrt{\epsilon_{\theta}{\rm D}_{F}(p_t|q_t)}$.
> We then discuss the bounds for these two terms separately. For the approximation error when learning the target vector field, we aim to minimize it through our loss function. As for the Fisher divergence, it is bounded under certain mild conditions according to Theorem 4.7 in [1]. Additionally, we highlighted that the upper bound of the Fisher divergence could be further optimized by applying high-order regularization terms, as shown in Theorem 3.2 in [2]. In this work, we only minimized the approximation error and found that it already brings strong empirical performance.
>
> As for Eq. (28), it is derived by substituting the bound for KL divergence from Eq. (27) into Pinsker’s inequality, $\operatorname{TV}(p_d(x|c),p_{\theta}(x|c)) \le \sqrt{1/2\operatorname{KL}(p_d(x|c)\mid p_{\theta}(x|c))}$, as shown in Eq. (25).
>
> We have revised and clarified these portions of the proof in our updated manuscript on page 15. We hope these adjustments provide a clearer understanding of this section.
>
> [1] Sitan Chen, Giannis Daras, and Alex Dimakis. Restoration-degradation beyond linear diffusions: A non-asymptotic analysis for ddim-type samplers. In International Conference on Machine Learning, pp. 4462–4484. PMLR, 2023.
>
> [2] Cheng Lu, Kaiwen Zheng, Fan Bao, Jianfei Chen, Chongxuan Li, and Jun Zhu. Maximum likelihood training for score-based diffusion odes by high order denoising score matching. In International Conference on Machine Learning, pp. 14429–14460. PMLR, 2022.
>
>
>
>
> **Question 2: In the proof of the theorem, some Lipschitz assumptions are made. It would be good to make these explicit in the theorem statement, in which they are currently not mentioned.**
>
> **Response:**
> We appreciate your feedback and attention to detail. Indeed, the Lipschitz conditions for the target and NN-approximated vector field are pivotal for the bound of the KL divergence in Eq. (27). Accordingly, we have included the Lipschitz conditions in the statement of Theorem 1 on page 7.
>
>
>
>
>
> **Question 3: The step from equation 31 to equation 32 could be explained in a bit more detail.**
>
> **Response:**
> Thank you for your feedback. We have revised and clarified these parts of the proof in our updated manuscript on page 16.
>
> We also clarify the transition from Eq. (31) to Eq. (32) as follows:
>
> First, in Eq. (30), the term $\|v _{\theta}(x,t,c)-\hat{v} _{\theta}(x,t,c)\|$ represents the error between the continuous and discretized vector fields. We then introduce an auxiliary term, $v _{\theta} (x,\lfloor tk \rfloor/k,c)$, in Eq. (31). This term represents the vector field at the previous discretized time point.
>
> Next, we apply the triangle inequality to Eq. (31).
>
> For the first term, $\|v _{\theta}(x,t,c) - v _{\theta}(x,\lfloor tk \rfloor/k,c)\|$, we consider the Lipschitz condition of the Neural Network (NN)-approximated vector field with respect to t and derive an upper bound, $L^{t} _{v _{\theta}}\frac{1}{k}$, as shown in Eq. (32).
>
> For the second term, $\|v _{\theta}(x,\lfloor tk \rfloor/k,c) - \hat{v} _{\theta}(x,t,c)\|$, we replace the discretized vector field $\hat{v} _{\theta}(x,t,c)$ with the the one on the previous discretized time point, as $\hat{v} _{\theta}(x-\hat{v} _{\theta}(x,t,c)(t-\lfloor tk \rfloor/k),\lfloor tk \rfloor/k,c)$.
> Under the Euler discretization methods, these two vectors are identical within the interval $[\lfloor tk \rfloor/k,t]$. We then consider the Lipschitz condition of the NN-approximated vector field with respect to x and derive an upper bound, $L ^{x} _{v _{\theta}}\hat{v}
>  _{\theta}(x,t,c) \frac{1}{k}$, as shown in Eq. (32).
>
> We hope these adjustments provide a clearer understanding of this section, and we look forward to hearing any further comments or questions you might have.

---

> > ### Comment · Reviewer_85JM · 2023-11-22
> > **Answer to Author Response**
> >
> > Thank you for answering to my questions, the changes in the revision clarify the problematic parts. After reading the other reviews I see that most of the reviewers shared my concern of limitations in the diversity of the experiments. Therefore I appreciate the authors adding the additional two experiments on more applied problems, which also show strong performance of the proposed method. In my opinion, this significantly improves the experiment section and I remain with my initial recommendation of accepting the paper.

---

> > > ### Author Response · Authors · 2023-11-22
> > > **Response to the comment**
> > >
> > > Dear Reviewer 85JM,
> > >
> > > We appreciate your time in re-evaluating our manuscript and are pleased to hear that our revisions have clarified the areas of concern. Your feedback has been invaluable in refining our work.
> > >
> > > We are grateful for your continued support and your recommendation to accept the paper. Your constructive feedback has helped to enhance the quality of our research significantly.
> > >
> > > Thank you once again for your time and insights.
> > >
> > > Best Regards,
> > >
> > > The Authors

---

### Official Review · Reviewer_ibgc · 2023-10-31

**Soundness:** 4 excellent
**Presentation:** 3 good
**Contribution:** 4 excellent
**Rating:** 8
**Confidence:** 4

**Summary:**

This paper presents a learning-based method to compute solutions for multi-valued non-convex optimization problems. The method is based on RectFlow method.

**Strengths:**

The proposed method addressed an important optimization problem. The concept of instead of learning input-output mapping, training a model to learn input-output-distribution is a novel concept. The authors provided concrete theoretical framework, and reasonable amount of empirical evidence to support the claims. The writing of the manuscript is clear and easy to understand.

**Weaknesses:**

The sampling portion of the manuscript can be further improved. The authors imply that due to the inexact approximation of NN vector field, and the discretization error when solving the forward solution of the ODE, the final results may not be as optimal as theoretically shown. This is fully understandable, however I would expect the authors to provide a more clear explanation, and also provide empirical solutions to address them. In the current version, the authors seem only point out to possible solutions.

Secondly, the explanation of the experimental results needs to be clarified.

**Questions:**

1. Section 4.3, it is understandable that there are issues in forward ODE solve. The authors pointed out a few possible solutions. Are these methods used to generate the results reported in section 6?

2. I would like to see expanded text of Section 4.3, and some ablation study if there are multiple solutions.

3. Section 6.2, Table 3, what are the `speedup` column compared against?

---

> ### Author Response · Authors · 2023-11-17
> **Response to Reviewer ibgc**
>
> We thank the reviewer for the time and effort in reviewing our paper and the positive encouragement. We address the concerns one by one in the following.
>
> **Comment 1: The sampling portion of the manuscript can be further improved.**
>
> **Response:**
> Thanks for your feedback. We have re-written this subsection with a more organized explanation to better illustrate the sampling process and the procedures to mitigate optimality loss and ensure solution feasibility. See Sec. 4.3 on page 6 in the revised manuscript (uploaded).
>
> --------
>
> **Comment 2: The explanation of the experimental results needs to be clarified.**
>
> **Response:**
> Thanks for your feedback. We have clarified the basic experiment settings in Sec. 6 and included other details in Appendix C from page 17-20 in the revised manuscript (uploaded).
>
>
> --------
>
>
> **Question 1: Section 4.3, it is understandable that there are issues in forward ODE solve. The authors pointed out a few possible solutions. Are these methods used to generate the results reported in section 6?**
>
> **Response:**
> Thank you for your question. Indeed, we apply the techniques outlined in Section 4.3 in the experiments presented in Section 6.
>
> Specifically, we implemented the following strategies:
> - (i) Sample-Then-Selection Strategy: we used this approach to enhance the optimality of the solutions generated for the optimization problems tested in Section 6.
> - (ii) Greedy Decoding Strategy: This strategy was employed to recover feasible discrete solutions for the combinatorial problem presented in Section 6.
>
> These techniques were critical in ensuring the optimality and feasibility of our generated solutions, and we have clarified their usage in our revised manuscript. See Section 6 on page 9 in the updated manuscript.
>
> We hope this response addresses your question, and we look forward to any further queries you might have.
>
> --------
>
>
> **Question 2: I would like to see expanded text of Section 4.3, and some ablation study if there are multiple solutions.**
>
> **Response:**
> Thank you for your valuable feedback. We have expanded Section 4.3 in the revised manuscript, providing a more detailed explanation of the sampling process and the strategies we implemented to mitigate optimality loss and ensure the feasibility of solutions. You can find this expanded discussion on page 6 of the revised manuscript.
>
> In response to your suggestion on the ablation study, we conducted one focusing on the sampling-then-selection strategy. This study illustrates the optimality under varying numbers of sampled solutions and discretized ODE steps. You can find these results in Appendix C.5 on page 20 of the revised manuscript. We believe this ablation study not only illustrates the empirical performance under different parameter settings but also offers further insight into the performance guarantee presented in Theorem 1.
>
> We hope these revisions and additions address your queries adequately, and we look forward to any further feedback you might have.
>
> --------
>
>
> **Question 3: Section 6.2, Table 3, what are the speedup column compared against?**
>
> **Response:**
> Thank you for your question. In Table 3, the "Speedup" column is measured in comparison to traditional iterative solving methods, where we solve those problems with the Gurobi solver when the graph size is less than $100$ to find global optimal solutions.
> For a graph size greater or equal to $100$, we adopt the heuristics provided in NetworkX to find local optimal solutions.
> We have also specified those settings for data generation in Appendix C.4 on Page 20.

---

> ### Author Response · Authors · 2023-11-23
> **Gentle reminder**
>
> Dear Reviewer ibgc,
>
> We hope this message finds you well. We're reaching out to gently remind you that the deadline for our rebuttal process is imminent, due in just a few hours.
>
> In response to your insightful feedback, we have undertaken significant revisions to our manuscript, particularly within the solution generation section and experimental section. Your constructive comments have been pivotal in refining our work, and we are keenly interested in hearing your thoughts on these changes.
>
> We are fully aware of the demands on your time and deeply appreciate your dedication to this rigorous review process. Your contributions are crucial to the advancement of our research.
>
> Once again, we extend our sincere gratitude for your time and invaluable input.
>
> Best Regards,
> The Authors

---

### Official Review · Reviewer_rFky · 2023-11-01

**Soundness:** 2 fair
**Presentation:** 3 good
**Contribution:** 2 fair
**Rating:** 5
**Confidence:** 3

**Summary:**

The rectflow model is applied to approximate the one-to-many solution mapping function in this paper. Given the Boltzmann distribution assumpation, this work transforms RectFlow model (Liu et al., 2022) to solve (potentially) nonconvex optimization problems. The experiments showcase RectFlow is better than some other generative models rather than state-of-the-art models for each nonconvex problems.

**Strengths:**

Solution generation is well motivated in the paper and the background is clearly written. Despite the straightforward application of RectFlow, the effect of the method is manifested in experiments by evaluating on simple optimization problems. The optimality loss and runtime complexity analyzed to show the efficiency.

**Weaknesses:**

1) Experiments are too simple and only easy problems are tested. The first simple function estimation with a limited feasible interval is oversimplified. The second combinatorial optimization experiment uses three relatively simple problems that have simple constraints. In this sense, the constraint handling power and the mapping power under complex scenarios are not showcased. More complex problems with multiple constraints can be added to enhance the experimental part.
2) Many generative models are designed for optimization problems. Despite the comparison with basic generative models, the generative models for optimization problems should be compared. A lot of them learn solution distributions rather than single-valued mappings, which share the same motivation in this paper. I believe they can be easily used in the tested problems in this paper. Please refer to A GNN-guided predict-and-search framework for mixed-integer linear programming, DIFUSCO: Graph-based Diffusion Solvers for Combinatorial Optimization, SurCo: Learning Linear Surrogates For Combinatorial Nonlinear Optimization Problems
3) The method is too straightforward by applying RectFlow, lowering the novelty of this work. Although the complexity is analyzed, the empirical results do not support the theory that the method is able to solve complex problems with multiple constraints.

**Questions:**

1. Can the authors clarify the constraints of the problems used in experiments?
2. The authors claim nonconvex leads to more multiple-valued input solution associations. But convex problems like multi-solution TSP may also have many one-to-many mappinps. Is there any explainations on this viewpoint?
3. The authors claim "existing studies endeavor to identify a single possible mapping from the potential multiples". It is not totally correct as considerable research focuses on estimating solution distributions given instances and many diversity techniques are adopted to encourage models finding more near-optimal solutions.
4. Despite the analysis of complexity, did authors try to solve large or complex problems? Any empirical insights on performance of the method in problem scales and constraint handling?

---

> ### Author Response · Authors · 2023-11-17
> **Response to Comment 1 for Reviewer rFky**
>
> We thank the reviewer for the time and effort in reviewing our paper and the positive notes. We address the concerns one by one in the following.
>
> **Comment 1: Experiments are too simple and only easy problems are tested. The first simple function estimation with a limited feasible interval is oversimplified. The second combinatorial optimization experiment uses three relatively simple problems that have simple constraints.**
>
> **Response:**
> Thank you for your feedback. The experiments in the paper are actually carefully designed with clear purposes. As stated at the beginning of Sec. 6 in our submitted manuscript, our initial experiment was deliberately designed to visually illustrate the capability of various approaches in learning a complex multi-valued mapping, where each input has six valid outputs. We believe this facilitates an easier comparison of the learning capabilities across different methods.
>
> Subsequently, in our second experiment, we aimed to evaluate the performance of different frameworks on three classical combinatorial optimization problems in graph theory, which have widespread applications in various domains. Given that our approach is designed to solve continuous problems, we first transformed the combinatorial problems into their continuous counterparts. Then, we applied our method to acquire a quality solution, before utilizing rounding schemes to recover a feasible solution for the combinatorial problems. The results demonstrate a clear performance improvement of our approach over both state-of-the-art methods (including the diffusion model for combinatorial problems [1]) and plausible alternatives, as explicitly mentioned at the beginning of Section 6.
>
> Meanwhile, we also agree with your viewpoint that performance evaluation on larger and more complex problems could further enhance our paper. Consequently, we have conducted additional experiments for two real-world, non-convex problems featuring larger sizes and more complex problem structures.
> - The first is the **inverse kinematics problem**, which involves determining the optimal robot configuration for a given target position by solving a system of intricate non-linear equations.
> - The second is the **wireless power control problem**, which requires finding the optimal power level for a given transmission configuration by solving a non-convex optimization problem.
>
> Both problems demand real-time solutions, and the existence of multiple solutions presents a challenge to current neural network-based methods. The new experimental results are presented in **Section 6.2 on page 9** of the revised manuscript, and they further illustrate the effectiveness of our proposed method in solving non-convex problems with multi-valued mappings.
>
> [1] Zhiqing Sun and Yiming Yang. Difusco: Graph-based diffusion solvers for combinatorial optimization. arXiv preprint arXiv:2302.08224, 2023.

---

> ### Author Response · Authors · 2023-11-17
> **Response to Comment 2 for Reviewer rFky**
>
> **Comment 2: Many generative models are designed for (discrete) optimization problems. Despite the comparison with basic generative models, the generative models for optimization problems should be compared.**
>
> **Response:**
> Thank you for your comments. We would like to clarify that we are fully aware of the existing research on developing generative models for solving discrete and combinatorial problems. In fact, we have discussed several such works in Section 2 (Related Work) of our submitted manuscript, including the papers mentioned in your comments, e.g., DIFUSCO [1] and SurCo [2]. We have also implemented six state-of-the-art or conceivable alternatives, including the diffusion-based one proposed in DIFUSCO [1], and compared their performance with ours in the simulation section Sec. 6.
>
>
> Our work distinguishes itself from these existing studies by focusing on the development of generative schemes for solving continuous optimization problems, while the extant literature primarily addresses discrete and combinatorial problems. Existing schemes often leverage the unique discrete structures of the problems using methods such as attention mechanisms [3,4], which are not directly applicable to the continuous problems that our paper addresses. We will amplify this point in the introduction and related work sections of our revised manuscript.
>
> As far as we know, our work is the first to develop a generative learning framework for continuous (non-convex) problems with multi-valued mappings. This framework is designed to learn the input-to-solution distribution and use the learned mapping to generate near-optimal solutions for new inputs.
>
> We have conducted both theoretical analysis and numerical experiments to assess the performance of our proposed solutions, and we believe our contributions are unique. In terms of performance analysis, we have established the first results in the literature to analyze the probability of high-quality solutions when applying ODE-based generative models for optimization problems, and illustrate the tradeoff between optimality and run-time complexity.
>
> In our numerical experiments, we compared the performance of our scheme with state-of-the-art methods for several continuous optimization problems. This includes some methods adapted from those designed for solving discrete problems, including DIFUSCO as referenced in [1], as well as several conceivable alternatives not present in existing works. The results not only demonstrate our approach's effectiveness in efficiently solving non-convex problems with multi-valued input-solution mappings, addressing a significant research gap, but also suggest potential extensions to discrete settings.
>
> We trust that our revised manuscript will provide further clarity on these points, and we eagerly anticipate any additional feedback you may have.
>
>
>
> [1] Zhiqing Sun and Yiming Yang. Difusco: Graph-based diffusion solvers for combinatorial optimization. arXiv preprint arXiv:2302.08224, 2023.
>
> [2] Aaron M Ferber, Taoan Huang, Daochen Zha, Martin Schubert, Benoit Steiner, Bistra Dilkina, and Yuandong Tian. Surco: Learning linear surrogates for combinatorial nonlinear optimization problems. In International Conference on Machine Learning, pp. 10034–10052. PMLR, 2023.
>
> [3] Wouter Kool, Herke Van Hoof, and Max Welling. Attention, learn to solve routing problems! arXiv preprint arXiv:1803.08475, 2018.
>
> [4] Yeong-Dae Kwon, Jinho Choo, Byoungjip Kim, Iljoo Yoon, Youngjune Gwon, and Seungjai Min. Pomo: Policy optimization with multiple optima for reinforcement learning. Advances in Neural Information Processing Systems, 33:21188–21198, 2020.

---

> ### Author Response · Authors · 2023-11-17
> **Response to Comment 3 for Reviewer rFky**
>
> **Comment 3: The method is too straightforward by applying RectFlow, lowering the novelty of this work. Although the complexity is analyzed, the empirical results do not support the theory that the method is able to solve complex problems with multiple constraints.**
>
> **Response:**
> Thank you for the comment. We wish to emphasize that the central contribution of our work goes beyond the mere application of RectFlow to non-convex optimization. Our proposal is a methodological framework that leverages any generative model—RectFlow being an exemplary candidate—to learn the input-to-solution distribution mapping of a non-convex problem with a multi-valued input-solution mapping. The learned mappings are then utilized to generate near-optimal solutions for new inputs. To the best of our knowledge, our work marks the first attempt to develop a generative learning framework that addresses continuous (non-convex) problems with multi-valued mappings both rapidly and accurately, a challenge not easily surmounted by state-of-the-art neural network solvers for continuous problems.
>
> We would also like to highlight that we also make a unique contribution in Sec. 5.1, where we offer a performance analysis and guarantee on the likelihood of high-quality solutions when employing ODE-based generative models for optimization problems. We emphasize that the analysis techniques used are general and not specific to the RectFlow model. They can be applied to derive similar results for other generative models.
>
> Having addressed the potential misunderstandings above, we concur with your perspective that an evaluation of performance on complex problems with multiple constraints could provide further depth to our paper and bolster the theoretical framework. In response, we have conducted supplementary experiments on two real-world, non-convex problems with more intricate problem structures. Please refer to our response to **Comment 1** for more details. The fresh experimental results, showcased in **Section 6.2 on page 9** of the revised manuscript, further validate the effectiveness of our proposed method in tackling complex problems with multiple constraints.

---

> ### Author Response · Authors · 2023-11-17
> **Response to Question 1&2 for Reviewer rFky**
>
> **Question 1: Can the authors clarify the constraints of the problems used in experiments?**
>
> **Response:**
> Thank you for your feedback. We clarify the constraints in the non-convex problems featured in our experiments in the following. In short, the constraints for the optimization problem in the initial submission are linear constraints, but the number of constraints is actually abundant (as large as the square of the number of variables). The newly added experiments in the revised manuscript have either complex nonlinear constraints or non-convex objectives. We include the details as follows:
>
> - (i) Toy Example: we constructed a multi-valued mapping. For each input, the output of this mapping is constrained to lie among six valid points.
> - (ii) Combinatorial problems: as detailed in Appendix C.4 on page 19, maximum clique and maximum independent set problems have two types of constraints. The first one pertains to the adjacency of selected nodes in the graph, which are linear inequality constraints, and the number of them is as the number of edges in the graph. The second one is related to the discrete decision variable for each node in the graph. The maximum-cut problem only has a discrete decision variable for each node but with a non-convex objective function.
> - (iii) Inverse Kinematics Problem: in this problem, added in the revised manuscript and detailed in Appendix C.3 on page 19, we have a system of non-linear equations as constraints. These constraints establish the intricate relations between the target position and the robot arm configurations.
> - (iv) Wireless Power Control Problem: for this problem, also added in the revised manuscript and detailed in Appendix C.3 on page 19, we have constraints on the upper and lower bounds of the power control variables. The objective is a complex, non-convex function.
>
> We have included these clarifications in the revised manuscript to provide more detailed context for each problem set.
>
>
> **Question 2: The authors claim nonconvex leads to more multiple-valued input solution associations. But convex problems like multi-solution TSP may also have many one-to-many mappings. Is there any explanations on this viewpoint?**
>
> **Response:**
> Indeed, as you suggested, for non-strictly convex problems (like multipath utility maximization problem problems in network optimization), there can also be multi-valued input-solution mappings. Since our NN approach developed in this paper is for solving optimization problems with multi-valued input-solution mappings, it can indeed be applicable for solving non-strictly convex problems as well. We have included this point in the introduction in the revised manuscript.

---

> ### Author Response · Authors · 2023-11-17
> **Response to Question 3&4 for Reviewer rFky**
>
> **Question 3: The authors claim "existing studies endeavor to identify a single possible mapping from the potential multiples". It is not totally correct as considerable research focuses on estimating solution distributions given instances and many diversity techniques are adopted to encourage models finding more near-optimal solutions.**
>
> **Response:**
> Thank you for your feedback. We would like to clarify that we are fully aware of the existing research on estimating the solution distribution and sampling high-quality solutions from it. In fact, we have discussed several such works in Section 2 (Related Work) of our submitted manuscript.
>
> We made this claim under the setting of solving continuous optimization problems, where most existing focus is on identifying the input-to-solution mappings [1,2], as discussed in Section 2 (Related work). For those works focusing on estimating the solution distribution, also discussed in Section 2, most of them primarily are designed for discrete and combinatorial problems, where they often leverage the unique discrete structures of the problems using methods such as attention mechanisms [3,4], which are not directly applicable to the continuous problems that our paper addresses.
>
> To avoid confusion, we have modified this claim in the problem statement sections of our revised manuscript. We appreciate your insightful comment and look forward to further discussions.
>
> [1] James Kotary, Ferdinando Fioretto, and Pascal Van Hentenryck. Learning hard optimization problems: A data generation perspective. Advances in Neural Information Processing Systems, 34:24981–24992, 2021.
>
> [2] Yatin Nandwani, Deepanshu Jindal, Parag Singla, et al. Neural learning of one-of-many solutions for combinatorial problems in structured output spaces. In International Conference on Learning Representations, 2020
>
> [3] Wouter Kool, Herke Van Hoof, and Max Welling. Attention, learn to solve routing problems! In International Conference on Learning Representations, 2018.
>
> [4] Yeong-Dae Kwon, Jinho Choo, Byoungjip Kim, Iljoo Yoon, Youngjune Gwon, and Seungjai Min. Pomo: Policy optimization with multiple optima for reinforcement learning. Advances in Neural Information Processing Systems, 33:21188–21198, 2020.
>
>
> **Question 4: Despite the analysis of complexity, did authors try to solve large or complex problems? Any empirical insights on performance of the method in problem scales and constraint handling?**
>
> **Response:**
> Thank you for your insightful query regarding our framework's scalability and its ability to handle constraints. Indeed, we have put our framework to the test on sizable and complex problems:
>
> - In the initially submitted manuscript, we applied our framework to a combinatorial problem over a randomly sampled 100-node graph, corresponding to an input size of a $100^2$ adjacency matrix. Empirical evidence showed that our method's loss of optimality was negligible compared to other Neural Network-based approaches, even at this large scale.
>
> - In the revised manuscript, we evaluated our framework on the inverse kinematics problem, which encompasses complex non-linear equation constraints. Despite these complexities, our generated solutions persistently exhibited minor constraint violations compared to other NN-based approaches.
>
> An additional empirical observation worth noting is that increasing the number of sampled solutions and subsequently selecting the optimal one can significantly elevate the solution's optimality. This observation is further elucidated in the performance analysis presented in Theorem 1.
>
> These experiments and insights have been incorporated into the revised version of the manuscript. We eagerly anticipate your additional comments and suggestions.

---

> ### Author Response · Authors · 2023-11-22
> **Gentle reminder**
>
> Dear Reviewer rFky,
>
> We hope this message finds you well. We write to kindly remind you of the impending deadline for our rebuttal process, which is due in one day.
>
> In response to your valuable feedback, we have conducted significant revisions to our manuscript, particularly within the experimental section and our discussion in relation to existing generative approaches. Your insightful comments have been instrumental in refining the quality of our work, and we are keen to hear your thoughts on these modifications.
>
> We fully understand the demands on your time and deeply appreciate your commitment to this rigorous review process. Your contributions are essential to the enhancement of our research.
>
> Once again, we would like to express our gratitude for your time and invaluable feedback.
>
> Best Regards,
> The Authors

---

> ### Author Response · Authors · 2023-11-23
> **Gentle reminder**
>
> Dear Reviewer rFky,
>
> We hope this message finds you well. As the deadline for our rebuttal process is quickly approaching and is due in a few hours, we wanted to send a gentle reminder.
>
> In response to your insightful feedback, we have undertaken significant revisions to our manuscript, notably within the experimental section and in relation to our discussion of existing generative approaches. Your comments have been instrumental in refining our work, and we are eager to hear your thoughts on these modifications.
>
> We fully acknowledge the demands on your time and deeply appreciate your ongoing commitment to this rigorous review process. Your contributions are fundamental to the enhancement of our research.
>
> Once again, we extend our sincere gratitude for your time and invaluable feedback.
>
> Best Regards,
> The Authors

---

> ### Comment · Reviewer_rFky · 2023-12-05
> **Response**
>
> Thanks for the rebuttal. I increase my score a bit. But the novelty is not significant from my view since the approach mainly depends on RectFlow. Theoretical parts are most from RectFlow. It is not clear how they are applied to other models.

---

### Official Review · Reviewer_GsRT · 2023-11-03

**Soundness:** 3 good
**Presentation:** 2 fair
**Contribution:** 3 good
**Rating:** 6
**Confidence:** 3

**Summary:**

The authors present a generative learning framework for the map between input and solutions for non-convex multi-input optimization problems, which has a wide range of applications in science and engineering. This framework leverages a generative model to learn the
mapping from input to a solution of distributions.  A benefit of this framework is when we sample from the input-dependent solution distribution the corresponding sample with the highest probability should correspond to a feasible solution. Lastly, the authors theoretically characterize optimally the characterized solution and empirically show RectFlow outperforms other generative models in solving this task.

**Strengths:**

The strengths of the paper are:
* Finding a novel application for the RectFlow model.
* Theoretical contribution- Providing a statement about the optimally gap decreasing as the number samples increase from the learned distribution.
* Solving an important problem that has a lot applications in engineering and science domains.
* The experiments that were provided have strong empirical evidence the proposed method is better than other generative models and baselines.

**Weaknesses:**

The paper could improve improve in the following ways:
* The motivation of the problem formulation could be more clear because there is a lack of context on the benefits of solving the problem in a generative framework. It is unclear why the authors chose to solve these particular types of non-convex problems with a generative model and why practitioners should adopt this framework.
* The experimental section lacks plethora of empirical evidence, the proposed method is demonstrated on two toy problems. There is no evidence provided the proposed method would help you solve a particular engineering problem and perform better than current methodologies.
     * Mirror weakness- Figure 3 would be more clear if colors associated with the samples of the probability were given as heat map.

**Questions:**

1). What are the benefits of using a generative framework to solve these types of optimization problems?

2). Could the authors provide a framework for practitioners on how to adopt this methodology? What requirements are needed?

3). Could the authors provide the probabilities of the samples in Figure 3?

---

> ### Author Response · Authors · 2023-11-17
> **Response to Comment 1&2 for Reviewer GsRT**
>
> We thank the reviewer for the time and effort in reviewing our paper and the positive notes. We address the concerns one by one in the following.
>
> **Comment 1: The motivation of the problem formulation could be more clear.**
>
> **Response:**
> Thank you for your feedback. We chose the problem formulation defined in Eq. (1) due to its wide-ranging applicability to cover an extensive array of engineering applications, including continuous non-convex and relaxed combinatorial problems. The formulation's universal nature equips us to tackle diverse complex scenarios, hence enhancing the versatility and applicability of our approach.
>
> The development of the generative framework to solve the non-convex problem presented in Eq. (1) is guided by multiple factors that we have detailed in our manuscript's introduction.
> First, neural networks (NN) based methods have demonstrated promising potential in predicting high-quality solutions for constraint optimization problems with low run-time complexity. This is particularly beneficial in real-time operational contexts, where problems must be addressed repeatedly amid rapidly evolving parameters. For strictly convex problems, the input-to-solution mapping is well-defined, which allows efficient learning by NN.
>
> However, general non-convex problems often present multiple optimal solutions for identical inputs, indicating a complex, multi-valued input-solution mapping that poses challenges to existing NN-based methods in learning these mappings.
> To solve the fundamental issue associated with the multi-valued input-solution mapping, we develop the generative learning framework. It is designed to learn the input-dependent solution distribution for non-convex problems, thereby generating high-quality solutions with low runtime complexity, a critical aspect for real-time engineering operations.
>
> We have taken your feedback into account and clarified our motivation following the problem formulation in the revised manuscript uploaded.
>
> -------
>
> **Comment 2: The experimental section lacks a plethora of empirical evidence.**
>
> **Response:**
> Thanks for your comment. In the initial submitted manuscript, we have included three classical but non-trivial non-convex problems for the graph theory to test the generative learning framework. Those problems exhibit many symmetric solutions and pose a challenge to existing NN-based approaches, making them proper benchmarks as non-convex problems with multi-valued solution mapping to test our framework.
>
> We also agree the performance evaluation over more diverse and sophisticated engineering problems can further strengthen our paper. To this end, in the uploaded revised manuscript, we have conducted additional experiments for two non-convex problems featuring more complex problem structures.
> - The first is the **inverse kinematics problem**, which involves determining the optimal robot configuration for a given target position by solving a system of intricate non-linear equations.
> - The second is the **wireless power control problem**, which requires finding the optimal power level for a given transmission configuration by solving a non-convex optimization problem.
>
> Both problems demand real-time solutions, and the existence of multiple solutions presents a challenge to current NN-based methods. The new experimental results are presented in **Section 6.2 on page 9** of the revised manuscript, and they further illustrate the effectiveness of our proposed method in solving real-world non-convex problems with multi-valued mappings.

---

> ### Author Response · Authors · 2023-11-17
> **Response to Question 1 for Reviewer GsRT**
>
> **Question 1: What are the benefits of using a generative framework to solve these types of optimization problems?**
>
> **Response:**
> We appreciate your insightful feedback and would like to provide some additional context, as stated in the abstract and introduction, before addressing your question. There has recently been a surge in the development of neural network (NN) schemes to solve continuous optimization problems. The concept is to utilize the universal approximation capability of the NN to learn the input-solution mapping of a problem, allowing new inputs to be passed through the learned mapping to achieve a high-quality solution significantly faster than traditional iterative solvers. This addresses the urgent need for fast and accurate optimization problem-solving in various engineering domains. These NN methods perform well for problems with single-valued input-solution mappings, such as strictly convex problems. However, they struggle with problems that have multi-valued input-solution mappings, such as non-convex problems. We provide an illustrative example in Fig. 1 of the initial submission (page 2). Some methods have attempted to address these issues, but they either incur high complexity or lack performance guarantees, as discussed in the related work in Section 2 (pages 2 and 3 in the initial submission). It remains a significant challenge to efficiently solve non-convex continuous optimization problems using NN schemes, which is the focus of our paper.
>
> To address your question, to the best of our knowledge, our generative framework is the first to learn the input-to-solution distribution mapping for non-convex continuous problems (or non-strictly convex problems) and employ the learned mapping to generate quality solutions for new inputs. This approach avoids the complexities of learning multi-valued input-solution mappings and has several key features that make it attractive to both researchers and engineers. Table 1 in the initial submission (page 3) provides a structured comparison of our approach with other state-of-the-art methods (on NN solving non-convex continuous problems). We also summarize the comparison below for easy reference:
>
> - (i) Wide-ranging applicability: The generative framework is applicable to a broad range of engineering applications that can be formulated in (1).
>
> - (ii) High-Quality Solutions: The framework utilizes the universal approximation capability of the generative model to learn the solution distribution and can generate high-quality solutions from the learned distribution.
>
> - (iii) Low Runtime Complexity: Our framework has been optimized for efficiency, generating high-quality solutions with significantly lower runtime complexity than iterative solvers, making it particularly useful for real-time operations.
>
> - (iv) Performance guarantee: We provide theoretical guarantees for the optimality and complexity of our framework. This is an advantage over many existing NN-based approaches, which often lack such assurances, as discussed in the related work section.
>
> In conclusion, we believe our generative approach adds a valuable tool to the arsenal of developing NN schemes for solving general continuous optimization problems, and this paper represents the first step in this direction.

---

> ### Author Response · Authors · 2023-11-17
> **Response to Question 2&3 for Reviewer GsRT**
>
> **Question 2: Could the authors provide a framework for practitioners on how to adopt this methodology? What requirements are needed?**
>
> **Response:**
> We appreciate your valuable feedback. We summarize the framework outlining how practitioners can implement our proposed methodology, along with the necessary prerequisites as follows:
>
>
> **Offline training phase**:
> - (i) Data collection: We begin by assembling a training dataset that includes high-quality solutions tied to varying input parameters. This is accomplished by solving the target problem using an existing iterative solver.
> - (ii) Model training: With the data collected, we proceed to train a generative model to learn the input-dependent solution distributions. The process follows the loss function detailed in Section 4.2 of our manuscript.
>
> **Online inference phase**:
> - (i) Initial points sampling: Given a new input parameter, we sample a batch of initial points from a Gaussian distribution.
> - (ii) Candidate solutions generation: Next, we feed the sampled initial points into the trained generative model, which outputs a set of candidate solutions. These candidate solutions, by the design of our generative model, are samples from the learned input-dependent solution distribution.
> - (iii) Best solution selection: Finally, we assess the set of candidate solutions based on objective value and constraint violation, selecting the best one as the final output.
>
> We provide a guarantee of performance on the optimality of the final output and the run-time complexity of the online inference in Section 5.1 of the initial submission (now on page 7 in the revised manuscript). We also discuss the offline training complexity in Section 5.2.
>
> As for the requirements, as detailed in the procedures above, they include (i) a dataset composed of high-quality solutions and their corresponding input parameters and (ii) a generative model designed to approximate input-dependent solution distributions.
>
> We have expanded on these procedures in the revised manuscript for further clarity.
>
> --------
>
> **Question 3: Could the authors provide the probabilities of the samples in Figure 3?**
>
> **Response:**
> Thank you for the suggestion. In response, we have added a color bar to Figure 3 in the revised manuscript. This color bar represents the probability density for the generated solutions. Please see page 8 in the updated manuscript for reference.

---

> ### Author Response · Authors · 2023-11-22
> **Gentle reminder**
>
> Dear Reviewer GsRT,
>
> We hope this message finds you well. We wanted to gently remind you that we are approaching the deadline for our rebuttal process, which is in 1 day.
>
> We have taken your previous feedback into careful consideration and made substantial revisions to our manuscript, notably within the experiment section. We are eager to receive your insights on these changes, as your previous comments have been instrumental in enhancing the quality of our work.
>
> We understand the demands on your time and genuinely appreciate your commitment to this rigorous review process. Your contributions are pivotal to the improvement of our research.
>
> Thank you once again for your valuable time and feedback.
>
> Best,
> Authors

---

> ### Comment · Reviewer_GsRT · 2023-11-23
> **Response to the Authors**
>
> I want to thank the authors for addressing my concerns. After reading the revision I agree with Reviewer 85JM, that adding two additional experiments significantly improves empirical evidence of the proposed method. As a result, I will increase my score accordingly.

---

> > ### Author Response · Authors · 2023-11-23
> > **Response to the comment**
> >
> > Dear Reviewer GsRT,
> >
> > We appreciate your time and effort in re-evaluating our manuscript. We are pleased to know that our revisions, and particularly the addition of two more experiments, have addressed your concerns.
> >
> > Your feedback has been instrumental in refining our research, and we sincerely thank you for your constructive insights.
> >
> > Best Regards,
> >
> > The Authors

---

### Author Response · Authors · 2023-11-21
**Gentle reminder**

Dear Reviewers,

We hope this message finds you well. We wanted to gently remind you that we are approaching the deadline for our rebuttal process, which is in two days. We have made significant revisions to our paper based on your valuable feedback and are eager to hear your thoughts on these changes.


Your insights have been instrumental in improving the quality of our work, and we would greatly appreciate your further comments to ensure we have adequately addressed all your concerns.


We understand that you are busy, and we genuinely appreciate the time and effort you are investing in this review process.


Thank you once again for your time and effort!

---

### Meta-Review · Area_Chair_RHDy · 2023-12-05

**Metareview:**

General non-convex problems often yield multiple optimal solutions for the same inputs, indicating a complex, multi-valued input-solution relationship. Traditional learning methods, which are usually designed for single-valued mappings, face difficulties in training neural networks (NN) to effectively interpret multi-valued mappings, often resulting in suboptimal solutions. To tackle this challenge, the authors develop a generative learning approach that utilizes a rectified flow (RectFlow) model, grounded in ordinary differential equations. This approach focuses on learning the mapping from input to the distribution of solutions, leveraging the universal approximation capabilities of the RectFlow model. When confronted with a new input, the trained RectFlow model is utilized to sample high-quality solutions from the learned input-dependent distribution. The authors claim that this method surpasses comparable GAN and Diffusion models in terms of training stability and runtime complexity. The authors provide an analysis of the optimality loss and runtime complexity associated with their generative approach. They also carry out simulation results in solving non-convex problems proporting that their method achieves better solution optimality compared to recent NN strategies, while maintaining comparable feasibility and performance speed.

The reviewers overall had a positive assessment of the paper and liked the novel application of RectFlow, theoretical contribution, and the experimental results. They did raise some concerns about clarity and further evidence in the experiments. One reviewer raised concerns about the simplicity of the experiments and the novelty of the paper. The authors rebuttal seems to have addressed most of the concerns. Therefore, I recommend acceptance but I recommend the authors to further revise the paper according to the feedback of the reviewers.

**Justification For Why Not Higher Score:**

The paper has some nice ideas but it is clear from the reviewer comments that the paper is still a bit lacking on more convincing experimental results.

**Justification For Why Not Lower Score:**

average score 6.75 with only one score below 6. Paper also clearly has interesting ideas.

---

### Decision · Program_Chairs · 2024-01-16

Accept (poster)